



# Hourly surface meltwater routing for a Greenlandic supraglacial catchment across hillslopes and through a dense topological channel network

**Colin J. Gleason**[1], **Kang Yang**[2], **Dongmei Feng**[1], **Laurence C. Smith**[3,4], **Kai Liu**[5], **Lincoln H. Pitcher**[6], **Vena W. Chu**[7], **Matthew G. Cooper**[8], **Brandon T. Overstreet**[9], **Asa K. Rennermalm**[10], and **Jonathan C. Ryan**[3]

[1]Department of Civil and Environmental Engineering, University of Massachusetts Amherst, Amherst, 01002, USA
[2]School of Geography and Oceanographic CE1 Science, Nanjing University, Nanjing, 210023, China
[3]Institute at Brown for Environment and Society, Brown University, Providence, Rhode Island, 02912, USA
[4]Department of Earth, Environmental, and Planetary Sciences, Brown University, Providence, Rhode Island, 02912, USA
[5]Nanjing Institute of Geography & Limnology, CE2 Chinese Academy of Sciences, Nanjing, 210008, China
[6]Cooperative Institute for Research in Environmental Sciences (CIRES), University of Colorado Boulder, Boulder, CO, USA
[7]Department of Geography, University of California Santa Barbara, Santa Barbara, 93106, USA
[8]Department of Geography, University of California, Los Angeles, Los Angeles, CA, 90095, USA
[9]Department of Geology and Geophysics, University of Wyoming, Laramie, WY, 82070, USA
[10]Department of Geography, Rutgers, The State University of New Jersey, New Brunswick, NJ 08901, USA

**Correspondence:** Colin J. Gleason (cjgleason@umass.edu)

**Abstract.** TS1 Recent work has identified complex perennial supraglacial stream and river CE3 networks in areas of the Greenland Ice Sheet (GrIS) ablation zone. Current surface mass balance (SMB) models appear to overestimate meltwater runoff in these networks compared to in-channel measurements of supraglacial discharge. Here, we constrain SMB models using the hillslope river routing model (HRR), a spatially explicit flow routing model used in terrestrial hydrology, in a 63 km$^2$ supraglacial river catchment in southwest Greenland. HRR conserves water mass and momentum and explicitly accounts for hillslope routing (i.e., flow over ice and/or firn on the GrIS), and we produce hourly flows for nearly 10 000 channels given inputs of an ice surface digital elevation model (DEM), a remotely sensed supraglacial channel network, SMB-modeled runoff, and an in situ discharge dataset used for calibration. Model calibration yields a Nash–Sutcliffe efficiency as high as 0.92 and physically realistic parameters. We confirm earlier assertions that SMB runoff exceeds the conserved mass of water measured in this catchment (by 12 %–59 %) and that large channels do not dewater overnight despite a diurnal shutdown of SMB runoff production. We further test hillslope routing and network density controls on channel discharge and conclude that explicitly including hillslope flow and routing runoff through a realistic fine-channel network (as opposed to excluding hillslope flow and using a coarse-channel network) produces the most accurate results. Modeling complex surface water processes is thus both possible and necessary to accurately simulate the timing and magnitude of supraglacial channel flows, and we highlight a need for additional in situ discharge datasets to better calibrate and apply this method elsewhere on the ice sheet.

## 1 Introduction

The study of supraglacial streams and rivers atop the Greenland Ice Sheet (GrIS) is an emerging subfield with implications for the physical understanding of ice sheet subglacial hydrologic systems, ice motion, and sea level rise (Irvine-Fynn et al., 2011; Rennermalm et al., 2013; Chu, 2014; Flowers, 2018; Pitcher and Smith, 2019). When the GrIS surface

melts, meltwater that is not evaporated, stored, or refrozen moves through what is now understood to be a complex perennial hydrologic system distinct from terrestrial hydrology (Yang et al., 2016; Pitcher and Smith, 2019). Recent advances in mapping (Lampkin and VanDerberg, 2014; Rippin et al., 2015; Smith et al., 2015, 2017; Yang and Smith, 2016), modeling (Banwell et al., 2012, 2016; Clason et al., 2015; Karlstrom and Yang, 2016; Yang et al., 2018) and measuring (McGrath et al., 2011; Legleiter et al., 2014; Gleason et al., 2016; Smith et al., 2017) supraglacial channel networks have revealed numerous similarities to terrestrial watersheds, but their scale and remoteness have limited the number of field studies.

This new appreciation for supraglacial hydrologic processes has emerged at a time of increasing accuracy and sophistication of surface mass balance (SMB) modeling of the GrIS. SMB models use regional atmospheric forcing to simulate GrIS surface mass balance components, including the amounts of meltwater production and of liquid water in excess of evaporation and retention and refreezing (termed "runoff") available for hydrologic functions (Fettweis et al., 2020; Vernon et al., 2013). SMB models here refer to any global and/or regional circulation model (G/RCM) or reanalysis that explicitly simulates ice sheet surface runoff. These models are grid-based and operate at pan-GrIS scales, producing a single runoff value for a given model grid and time step. Note that the terrestrial hydrology community commonly uses the term "water excess" to represent the volume of water available for routing after hydrologic processes, while the glaciology community uses the term "runoff" to represent this same quantity specific to ice sheets. Most existing SMB models do not route this runoff and instead assume that all runoff not refrozen in snow or firn leaves the ice sheet as soon as it is produced (Fettweis et al., 2020). In reality, observations of the GrIS surface indicate that lake impoundment (e.g., Arnold et al., 2014), flow through weathering crust (e.g., Cooper et al., 2018), and transport through supraglacial stream and river networks modify the timing and magnitude of excess water reaching moulins or the ice sheet edge (Smith et al., 2017). Modeling these processes is precisely analogous to the use of land surface models in terrestrial hydrology, whereby a land surface model (SMB model here) produces gridded water excess (runoff here) and then routes this water with a coupled routing model. Coupling surface water processes to SMB models, loosely or tightly, is thus needed for a fuller representation of GrIS supraglacial hydrology to align this field with practices in terrestrial hydrology (e.g. Bates et al., 1997; Beighley et al., 2009; Wood et al., 2011; Lin et al., 2019 TS2).

Previous studies have begun to stich these two research avenues together. For example, Banwell et al. (2012) used Darcy's law to describe meltwater flow routing through snow and Manning's equation to describe lateral runoff transport across bare ice and then later used this meltwater to fill supraglacial lakes or supply surface meltwater to moulins (Banwell et al., 2013, 2016). Leeson et al. (2012) similarly used Manning's equation to transport water in a 2D grid-based routing scheme, assigning all grids a uniform Manning's $n$ while not explicitly defining flow differences between flow in channels and flow over bare ice. Liston and Mernild (2012) also applied mass conservation at the grid cell level to route runoff between grid cells and did not account for the presence of channels that convey this runoff with distinct hydraulics. Smith et al. (2017) attempted to address this channel routing via the classic empirical Snyder synthetic unit hydrograph (SUH) model (Snyder, 1938) to calculate discharge hydrographs for the terminal moulins of 799 internally drained surface catchments in the southwest GrIS. Yang et al. (2018) used a similar classic empirical model, the rescaled width function (RWF; Rinaldo et al., 1995), to partition the ice surface into slow-flowing interfluvial (i.e., hillslope) and fast-flowing (open-channel) zones and calculated moulin discharge while improving the physical realism of the supraglacial routing process. Importantly, Yang et al. (2020) CE4 demonstrated the likelihood of subsurface unsaturated zone flow even through bare glacial ice, a phenomenon confirmed by field (Cooper et al., 2018; Irvine-Fynn et al., 2011; Munro, 2011) and theoretical (Karlstrom and Yang, 2016) studies. Yang et al. (2020) recently compared several of these empirical models and found they introduce significant variability in diurnal moulin discharges and corresponding subglacial effective pressures.

These previous efforts demonstrated successful meltwater transport modeling on the GrIS ablation zone and its necessity, but their relative simplicity allows space for the application of sophisticated routing models from terrestrial hydrology to be applied to ice sheet surfaces more generally. For instance, Lin et al. (2019) TS3 used gridded estimates of water excess (analogous to runoff) to simulate daily flows in nearly three million river reaches between 1979 and 2013 with fully conserved mass and momentum in realistic river networks globally. This undertaking was the first demonstration of this capability at global scale following years of well-established theoretical work and advances in hydrologic representation for big data. This routing approach is suitable for representing GrIS surface water transport processes as gridded runoff on ice sheets must be routed through supraglacial rivers, lakes, and hillslopes (which include firn atop the GrIS), as on land. Building and calibrating models to route water through landscapes and channel networks while obeying fundamental principles of mass and momentum conservation is an established practice in terrestrial hydrology that may readily be applied to ice sheet surfaces as well.

There are several barriers to applying such routing for the GrIS at the catchment scale. First, routing models require a well-defined channel network with explicit and continuous topology. There have been demonstrations of network mapping (Yang et al., 2016) and topology generation (King et al., 2016), but to our knowledge no automated, large-network-scale (i.e., catchments with thousands of channels or more)

coupled extraction and topological connection work exists for the GrIS. Existing terrestrial routing models like the hillslope river routing model (HRR; Beighley et al., 2009) stand ready to route runoff "off the shelf", yet these cannot be applied until a generalizable automated extraction and topological connection process is available. Applying a model such as HRR could also further understanding of GrIS river networks, which is currently underdeveloped (Pitcher and Smith, 2019). For instance, the relative importance of hillslope flows and channel density on runoff transport have not been explored on a first-principles basis at network scales, and model parameters controlling hillslope friction, channel friction, and runoff reduction and augmentation could reveal how these physical processes interact to produce channel discharges.

In this paper we use HRR to advance the physical understanding of GrIS supraglacial meltwater transport processes as follows. (1) We automatically generate spatially explicit topological networks of varying drainage densities for a supraglacial catchment for which a brief (72 h) in situ record of outlet channel discharge is available. (2) We route water runoff generated by four different SMB models through these networks at an hourly timescale. (3) We constrain and calibrate the routing via hourly in situ discharge measurements and previously published field measurements of supraglacial channel frictions and velocities. Our initial routing results immediately revealed a mismatch between modeled and routed runoff and measured channel flows, so our philosophy for this study is to assume that measured discharge at the outlet is correct and calibrate SMB runoff volumes and channel properties to match discharge observations as mediated through the physics of the routing model. (4) To advance understanding of hillslope processes and channel density on meltwater transport, we design an experiment to test how the representation of hillslope processes and network density (as derived by our automated network generation process) affects the routing model. We ultimately route meltwater through thousands of supraglacial channels every hour, and we solve (via conservation of mass and momentum inherent to routing) for the roles of channel friction, hillslope delay, and network density in controlling the magnitude and timing of water fluxes through supraglacial channels and ultimately moulin injection in our test watershed. These procedures and results form a blueprint for the general coupling of runoff modeling, water transport, and channel processes atop the GrIS.

## 2 Study area and data

We develop our routing model for Rio Behar, a previously studied, internally drained supraglacial river catchment in southwest Greenland. First introduced by Smith et al. (2017), the Rio Behar catchment is approximately 63 km$^2$ and centered at 67.04° N TS5 and 48.55° W with a highly developed perennial and well-drained supraglacial stream and/or river during peak flow periods of late summer. Smith et al. (2017) report that the basin elevation spanned approximately 1200–1400 m in 2015, with air temperatures in the summer measurement period ranging from −3 to 2 °C TS6 and net radiation ranging from approximately −100 to 300 W/m$^2$. Previous work in the basin includes (i) a comparison of SMB runoff and field-measured discharge using a simpler routing method (Smith et al., 2017), (ii) a study of subsurface water storage in bare-ice weathering crust (Cooper et al., 2018), (iii) albedo mapping (Ryan et al., 2017), and (iv) satellite and uncrewed aerial vehicle (UAV) remote sensing work to map the catchment's supraglacial channel network (Ryan et al., 2017; Yang et al., 2018). Readers are referred to these published works for more information on the physical setting of the basin. Here we use the Rio Behar specifically because it is the only known large GrIS supraglacial river catchment with an hourly in situ record of channel discharge (see Sect. 2.2). Other discharge records exist, as, for instance, McGrath et al. (2011) who provide hourly discharge records for a small (1.1 km$^2$) catchment, while Chandler et al. (2013) TS7 give hourly moulin (fed by a channel) discharge for another small catchment, but the size of Rio Behar and the wealth of previous work therein makes it an ideal setting for this study. Using high-resolution remote sensing, the watershed is delineated to an in situ streamflow measurement point (Sect. 2.2) that defines the outlet located less than 1 km upstream of the catchment's terminal moulin. Because all meltwater runoff passing out of our watershed penetrates the ice sheet via a moulin, accurate modeling of this water flux is important for studies of GrIS subglacial hydrology and ice dynamics (Chu, 2014; de Fleurian et al., 2016; Banwell et al., 2016; Flowers, 2018; Davison et al., 2019)

### 2.1 Remotely sensed and SMB model data

A high-resolution remotely sensed supraglacial stream network for the Rio Behar catchment, mapped from a 0.5 m resolution panchromatic WorldView-2 satellite image acquired on 18 July 2015, was obtained from Smith et al. (2017), and this scale is sufficient for capturing the smallest streams in this region (Yang et al., 2018). The stream network product of Smith et al. (2017) was combined with a seasonally simultaneous portion of the 2 m resolution ArcticDEM digital elevation model (DEM) obtained from the Polar Geospatial Center (Porter et al., 2018) to produce two distinct supraglacial stream networks, as described in Sect. 3.2. The ArcticDEM has been widely used in GrIS hydrology studies and performed reasonably well in representing drainage patterns in previous work (e.g., Moussavi et al., 2016; Pope et al. 2016; Yang et al., 2020).

GrIS runoff was simulated by four models (HIRHAM5, MAR3.6, RACMO2.3, and MERRA-2). Data and detailed descriptions of these SMB models are provided in Smith et al. (2017), but in brief each of these models solves a local

surface energy balance from meteorological forcing to produce some amount of runoff produced after physical processes of melting, condensation, retention, and refreezing. This excess water is spatially gridded, and for a given grid cell the models each produce hourly runoff, which we assume is topographically constrained and transported exclusively via surface/near-surface transport. We take the average runoff in all grid cells intersecting Rio Behar (ranging from one to eight SMB grid cells for the four models) to arrive at a single hourly runoff value for each SMB model following Smith et al. (2017). We therefore have four different runoff forcings available for routing that cover from 1 month before the in situ measurement period through the end of the measurements (Sect. 2.2). Our goal for this paper is not to interrogate these models. Rather, we hope to highlight the nuances of supraglacial meltwater routing across a range of forcings.

## 2.2   In situ data

Two sources of field data are available for this study. The first source is an hourly acoustic Doppler current profiler (ADCP) discharge record published by Smith et al. (2017). An ADCP is an instrument that measures river flow depth via sonar ranging and vertical velocity profiles using Doppler shifts in the water column. The instrument is transited orthogonal to flow and makes its measurements in discrete bins which are then summed to arrive at the mass flux of water in the channel. ADCP outputs are thus correctly labeled as "estimates" of discharge rather than "measurements" as the measured quantities are depth and velocity and discharge is derived. However, the ADCP provides the most trusted and accurate method for estimating discharge used in hydrology, and its discharge estimates are frequently labeled as measurements (Gleason and Durand, 2020). Further reading on ADCP estimates of discharge and measurement protocols can be found in Turnipseed and Sauer (2010).

Smith et al. (2017) obtained hourly measurements of discharge via ADCP at the outlet of Rio Behar from 13:00 UTC on 20 July 2015 to 12:00 UTC on 23 July 2015. Smith et al. (2017) give a detailed description of measurement protocol for collecting and processing these ADCP discharges, and readers are referred to that publication for more information. ADCP estimated discharges ranged from 4 to 26 m$^3$/s, revealing that large supraglacial rivers do not dewater at night and can sustain peak flows comparable to streams of moderate catchment size in terrestrial hydrology. These ADCP discharges form the core HRR model calibration dataset for our study.

The second source of in situ data used here is a broad set of observations of supraglacial channel hydraulics collected in summer 2012 across 64 supraglacial streams and rivers of the southwest GrIS (Gleason et al., 2016). These in situ measurements consist of instantaneous supraglacial channel flow widths, depths, water surface slopes, and velocities collected using traditional surveying, radar velocimetry, and an ADCP. These measurements in turn yielded derivative estimates of discharge, stream power, Froude number (a classic index of flow velocity in open-channel hydraulics), and roughness coefficient (Manning's $n$) at 64 sites, representing the largest known empirical dataset of supraglacial channel hydraulic properties currently available in the literature. Site locations ranged from 502 to 1485 m elevation and up to 74 km inland from the ice margin, and instantaneous discharges ranged from 0.006 to 23.12 m$^3$/s in actively flowing channels 0.20 to 20.62 m wide. These observations are used to constrain our modeled roughness coefficients to produce realistic parameters and velocities. Section 3.3 describes this process fully. Note that we cannot use these observations to validate our routing model and instead use them to inform it. These point measurements could in theory be reproduced by our hydraulic model, but to do so would require measurements of channel properties and runoff upstream of each point for several hours/days before each hydraulic measurement was taken, and such data do not exist.

## 3   Methods

### 3.1   Experiment design

Our overall goal for this study is to improve the current understanding of supraglacial hydrological transport processes by classically modeling hillslope and channel routing. We test two experimental settings (inclusion/exclusion of hillslope flow, coarse-/fine-channel network densities) on four different SMB models to produce 16 experimental runs (4 runs per model; Fig. 1). These runs are labeled as either "fine" or "coarse" and "hillslope" or "non-hillslope"; so, for example, an experiment using a fine-network density and excluding hillslope processes would be labeled "non-hillslope fine." For each run, we calibrate 11 parameters: a global runoff correction coefficient (1 parameter), a spatially explicit channel roughness coefficient binned by channel slope (9 parameters), and a global hillslope roughness coefficient (1 parameter) to optimize modeled and measured discharge at the basin outlet (Sect. 3.3.2 gives full details). Model calibration statistics were used as indicators of the physical realism of each experiment, and we seek to identify robust, cross-SMB model parameter trends in our factorial experimental setting. Thus, we calibrate HRR 16 separate times to produce a set of results that vary by runoff forcing, channel density, and inclusion/exclusion of hillslope process.

Note that in all configurations (Fig. 1), we calibrate a runoff correction coefficient ($R_{coef}$). Previous work comparing SMB runoff to ADCP discharge at our field site reveals that the SMB runoff is frequently greater than observed discharge leaving the watershed (Smith et al., 2017). We therefore created a multiplicative runoff correction coefficient to either reduce or augment SMB runoff that is calibrated

within HRR without changing the timing of production. Previous routing studies have forced model runoff to equal the cumulative measured river discharge before further routing (Smith et al., 2017), yet this restrictive assumption amounts to an empirical ad hoc mass conservation rather than explicitly relying on hillslope and channel mass and momentum conservation across thousands of channels. Thus, we calibrate $R_{coef}$, together with the traditional HRR parameters (i.e., channel and hillslope roughness coefficients; Table 1; Sect. 3.3.2), for each model run to learn the total volume of excess needed in each case to simultaneously match both hydrograph timing and mass conservation. This allows our results and routing framework to guide our conclusions on the total volume of water needed to generate the outlet hydrograph as this volume might differ between network and hillslope configurations. Further, the use of a single $R_{coef}$ allows us to accurately model discharge without allowing the attribution of errors in runoff production: these could stem from SMB errors, unaccounted for refreezing, storage, or lake filling, surface transport that violates topographic constraints, englacial draining, or ADCP measurement error. Our framework is unable to apportion any gaps in runoff production and routed discharge to any of these sources, and thus our treatment of runoff as a bulk reduction/augmentation is faithful to our experiment design and article goals.

## 3.2 River network extraction

Although Smith et al. (2017) provide a topologically connected channel network for our study area (i.e., they explicitly defined how every channel is connected to every other channel throughout the entire network to allow water to flow from the headwaters to the outlet to obey observed channel connections), we are interested in generalizing the process of water routing in cases when preexisting channel network maps do not exist. Further, we must generate different river networks to test the effects of network density on the routing model. Therefore, we introduce a process to create models of complete river networks as defined by topography that can in theory be applied to any area of the GrIS with a high-quality DEM and a remotely sensed image. This topographically defined flow is a classic practice in terrestrial hydrology, and since all open-channel flow is gravity-driven, this practice applies for flow routing through any medium without substantial pressure forces. Topographically defined flow has therefore been applied/invoked for a variety of surfaces, including Mars (e.g., Dohm et al., 2001; Rodriguez et al., 2005; Fassett and Head, 2008)

To generate our river networks, (1) we first "burned" (i.e., lowered the pixel elevations) the remotely sensed stream map of Smith et al. (2017) into ArcticDEM, a standard hydrologic practice (e.g., Lindsay, 2016). This process ensures that channels are lower than surrounding topography as remotely sensed DEMs cannot "see" channel bottoms and therefore create smooth surfaces where surface water features exist.

Since we know that a river or stream channel is abruptly deeper than its surrounding banks, artificially lowering elevations where we observe channels ensures that these locations are the lowest feature in the surrounding terrain and therefore collect topographically driven water. In DEM processing for a hydrology, a depression is an area where water pools as the flow direction is always downhill as in the sides of a bowl. These depressions typically need to be artificially "filled", that is, their elevations need to be raised, as otherwise the topography indicates that water cannot leave once it enters the depression. Because we "burn in" stream locations to the DEM, standard sink filling is not required for this analysis (we lower streams rather than raise depressions), but two large topographic depressions in the DEM of our catchment required further processing even after burning in streams. Standard DEM preparation for network generation dictates that upstream depressions are filled, while outlet depressions are preserved, yet this assumption generated unrealistic parallel drainage channels upstream and no channels in the outlet depression for our data. To address this problem, (2) a priority-flood algorithm (Lindsay, 2016) was applied to breach the two depressions and to create a continuously flowing, realistic drainage network for the catchment (Fig. 2). Finally, (3) the parameter that drives network generation and ultimately channel density is the channel initiation threshold: the minimum area needed to form a free-flowing channel. This concept stems from the fact that above river headwaters, water simply flows through the soil and not on the surface until the water table elevation exceeds the soil elevation in a spring. We observe an exact analogue on the GrIS: channels dwindle in size until they become indistinguishable from wet firn and/or ice near topographic divides (Gleason et al., 2016). To estimate the impact of drainage patterns on meltwater routing, we tested both a large ($10^4 \, m^2$) and a small ($10^3 \, m^2$) channel initiation threshold to create a "coarse" and a "fine" supraglacial drainage network, respectively, from the DEM (Fig. 2). These two modeled stream networks both follow the channel map from Smith et al. (2017), with the key difference that the coarse network does not produce the narrowest streams we know to exist. This enabled us to test the effects of including or excluding very small tributary streams on surface water routing. We assign channel widths to each DEM-derived channel from the channel map of Smith et al. (2017), and since the DEM process begins with burning in these streams, there is always a 1 : 1 assignment of channel width from imagery to network model. Our fine-channel network produces streams with a minimum width of 0.5 m, matching to the correct order of magnitude the reporting by Gleason et al. (2016) of channels as narrow as 0.2 m. The coarse network produced streams with a minimum width of 0.7 m, suggesting it is excluding the smallest streams in the remotely sensed map. GrIS supraglacial channels incise and meander over time, yet HRR cannot represent this behavior and instead assumes that channels remain fixed in space and time. It would be possible to derive expected erosion and

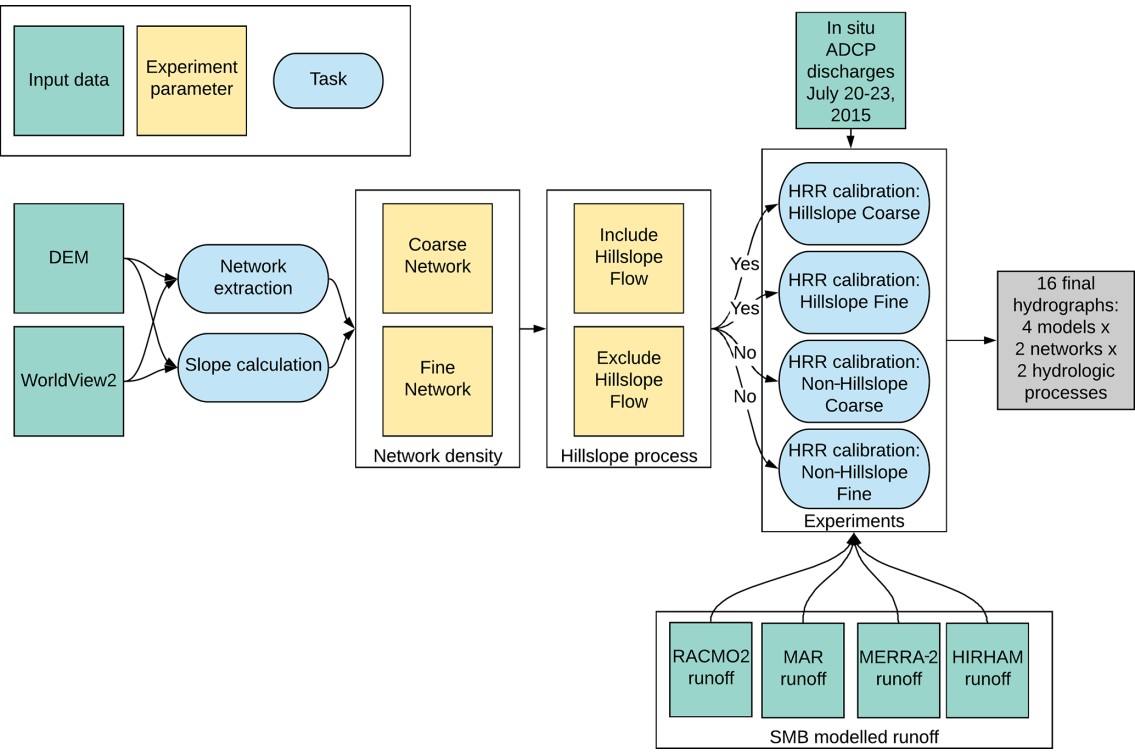

**Figure 1.** Schematic diagram of our experimental design and modeling procedure. Hillslope river routing (HRR) model inputs, processes, and outputs are labeled. This workflow yields 16 independent hydrographs by considering fine vs. coarse supraglacial channel network densities and inclusion vs. exclusion of hillslopes in addition to open-channel flow.

incision (and additional meltwater) due to frictional heating of the channels, but without including a radiation budget and ice property data we could not model how the stream network changes in time nor satisfactorily model this additional meltwater with commensurate sophistication to the SMB runoff forcing (i.e., tight coupling with SMB models). Instead, we model these network snapshots with HRR being as loosely coupled with SMB runoff (as opposed to tightly coupled, when SMB runoff would be an input into network generation), which is reasonable for our 1-month experiment (Sect. 3.3.1).

Our river network extraction produced two topologically connected networks of 1044 and 8095 channels (coarse and fine, respectively; Fig. 2). The coarse network has six stream orders (the smallest streams on the landscape are defined as order 1, and every junction of stream produces a new stream of higher order) and the fine network seven orders. Stream orders are a shorthand for the hydraulic complexity of a network as the number and length of streams in a given order both increase geometrically (Horton, 1945). Therefore, our finding of almost an order of magnitude more channels in the fine seven-order network than the course six-order network matches theory. The networks are topologically complete (i.e., all channels are explicitly connected to one another and preserve their hydrologic hierarchy), allowing for successful routing without the need for further correction of

network connections. The main trunk streams only are visible in the coarse network, and lakes connected to the channel network (i.e., have an inflow and outflow) are represented by wide, shallow "throughflow" river segments as all are nonterminal with outflow channels. Lakes on the GrIS evolve seasonally; they begin pooling water in the early melt season until an outlet elevation is reached, and then they begin to spill downstream. Our data come from peak melt season when lakes are full, and thus any lake connected to the network will behave fluvially, that is, it will spill according to its slope, volume, and lateral input via the conservation of mass and momentum. Further, Fig. 2 indicates that there are likely no lakes in the watershed that are disconnected from the channel network – our drainage density is sufficient to ensure that lakes of any appreciable size would be captured as a throughflow segment.

### 3.3 River routing

#### 3.3.1 Model setup

HRR routes water excess over the land surface and through channels. In channels, it follows the Muskingum–Cunge equation, a kinematic wave approximation of the 1D St. Venant equations (conservation of mass and momentum in an open channel; Cunge, 1969). HRR uses an explicit kine-

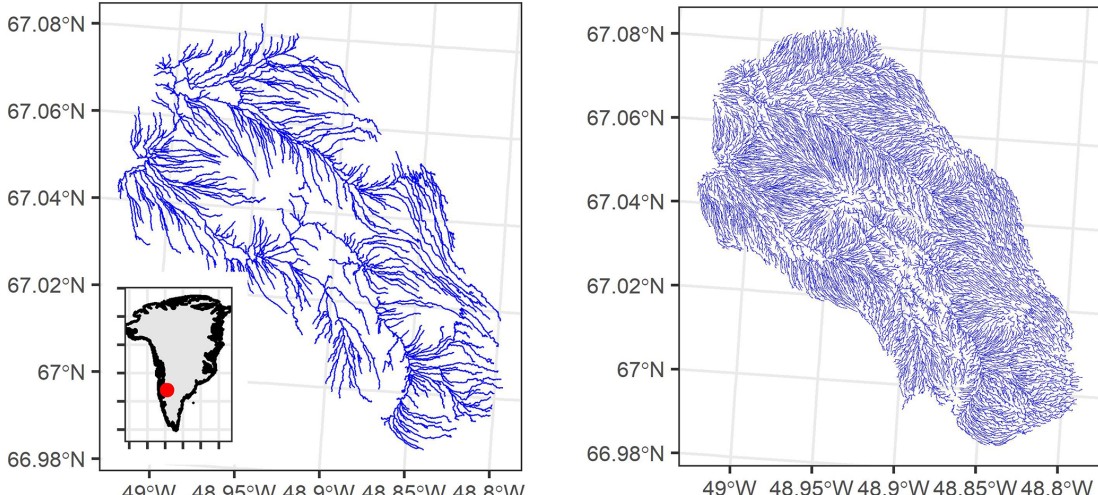

**Figure 2.** The 1044 segment "coarse" network and the 8095 segment "fine" network were automatically extracted from a DEM and remotely sensed data. These river networks represent different channelization area thresholds, and test how assumptions of network density control hydrologic process.

matic wave for hillslope transport as non-channelized overland flow (Li et al., 1975). HRR requires inputs of channel widths and lengths, which are assumed to be invariant and derived from remote sensing (Sect. 2.1), channel slope, and each channel's subcatchment area and total upstream area, as derived here from the DEM, in which bed slope is assumed to equal the free surface flow, consistent with Manning's equation. In addition, the network topology derived in Sect. 3.2 is required so that HRR can conserve mass and momentum in a downstream direction and across channel junctions. HRR is one of several routing models that classically conserve mass and momentum designed for large-network applications. Our choice of HRR is based on familiarity, model speed (written in FORTRAN and called `CE6` from the RStudio software package here), and its rigorous representation of network routing and classic open-channel flow hydraulics.

HRR routes time-varying runoff onto existing flows, commonly onto a baseflow in terrestrial hydrology. We "spin up" the model by routing a constant forcing of median observed ADCP flow through the model rather than attempt to define a minimum baseflow. This steady forcing allows all channels to fill with water and accurately transfer runoff from the SMB models through the system. We used a 3-month spin-up period then temporally varied flows beginning on 1 July from SMB forcing. Our experiment begins on 20 July, and thus the model has time to adjust to runoff forcing and mitigate the impact of this spin-up flow before we begin to validate the model.

### 3.3.2 Model calibration

Nearly all hydrologic models require calibration to function well. To calibrate terrestrial routing models, hydrologists typically iterate parameters until hydrographs at one or more

reaches match a stream gauge in that reach. Here, we have calibration data available only at the basin outlet, so we calibrate our routing model to outlet discharges despite producing discharges in thousands of reaches. A very large amount of the literature on hydrologic model optimization and calibration exists, and interested readers are referred to Kirshner `TS9` (2006) and Gupta et al. (1998) for broad overviews of the subject. We perform calibration using an established evolutionary algorithm (EA; NSGA II; Deb et al., 2002) as EAs are efficient estimators in large parameter spaces that can achieve near-optimal results (Gleason and Smith, 2014). This calibration ensures a heuristically optimized outlet hydrograph but does not explicitly calibrate upstream reaches. However, since outlet flows are the sum effect of the routing delays and volumes of all upstream reaches, and since we explicitly conserve mass and momentum, a well-calibrated outlet should satisfactorily model upstream flows, but we cannot validate these upstream reaches. Therefore, we constrain allowable parameters in upstream reaches (and therefore their discharges and velocities) using the in situ observations of Gleason et al. (2016).

We calibrate 11 constrained parameters (Table 1) which represent three physical concepts: channel friction (here expressed as Manning's $n$ and binned by upstream area into 9 separate parameters), hillslope friction, and a water excess adjustment coefficient. Channel friction is represented by Manning's $n$, and the EA solves for a single $n$ per bin and assigns that $n$ to all streams falling within that drainage area threshold. Manning (1891) generalized open-channel flow into a simple equation in which all flow resistances are lumped into a single empirical parameter $n$, and over a century of subsequent research has related $n$ to landscape variables, channel form, and other geomorphic controls. Our bin-

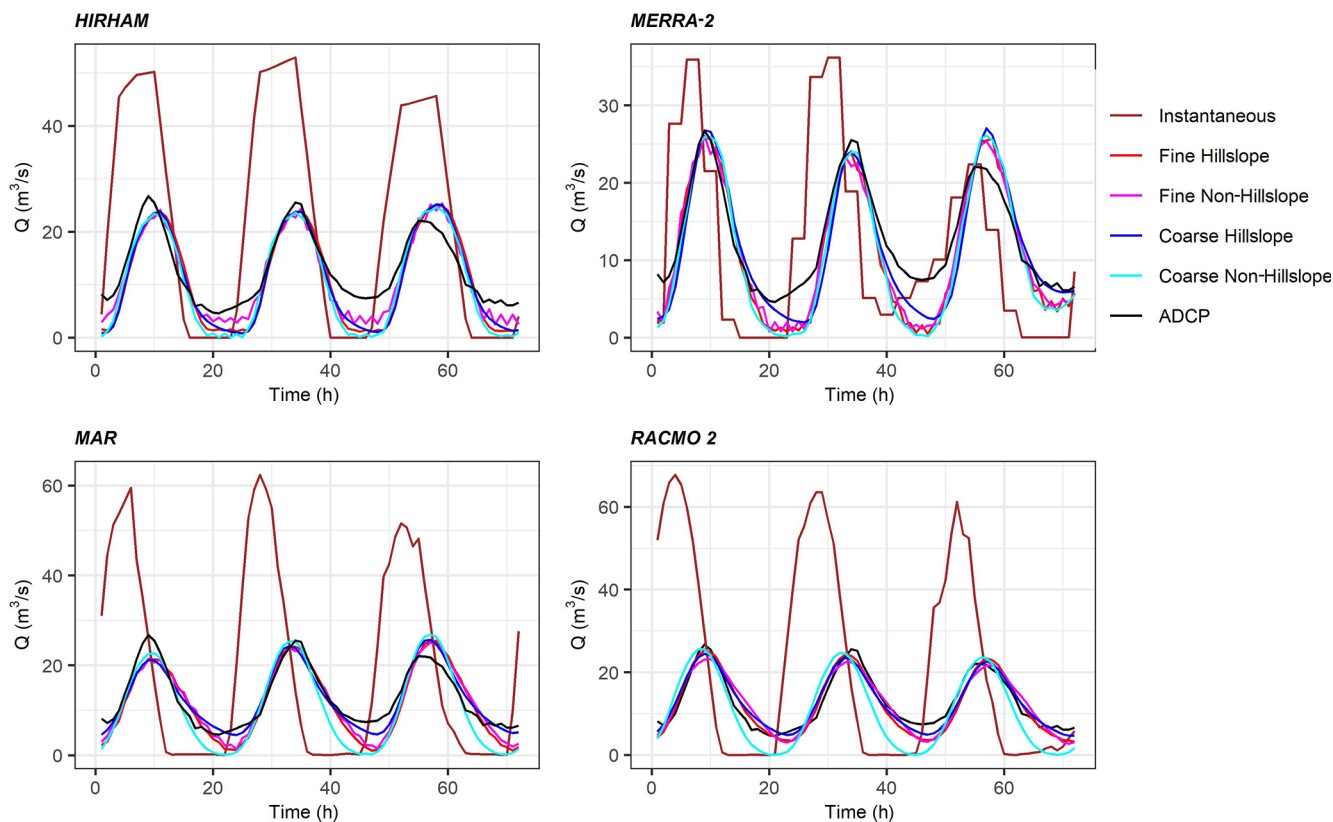

**Figure 3.** The hourly in situ ADCP hydrograph at the basin outlet (in black) clearly shows the necessity of delaying and reducing SMB modeled runoff ("instantaneous", brown lines) to match field observations. Even after coupling SMB models with HRR routing models, most simulations underpredict low flows. Peak flows are relatively well modeled, although ADCP peak recession is only modeled correctly by RACMO2-forced routing. TS8

ning of Manning's $n$ follows general hydraulic correlations between channel size, slope, total discharge, and $n$ (Brinkerhoff et al., 2019). Hillslope flow is modeled as an explicit kinematic wave for non-channelized flow (Li et al., 1975), which requires a surface roughness coefficient (i.e., hillslope friction), and we limit hillslope friction to between 0.05 (non-dimensional; a hillslope with friction equivalent to a rough channel) and 25 (a hillslope with extreme friction to approximate slow interflow through weathering crust). For context from the terrestrial hydrology literature, McCuen (1998) TS10 provides a reference table for watershed surface roughness with hillslope friction values ranging from 0.01 to 0.8. Kalyanapu et al. (2009) TS11 developed another reference table based on the National Land Cover Database, and their values range between 0.01 and 0.4, while Hergarten and Neugebauer (1997) suggest friction up to a value of 1. Thus, we allow GrIS ice surface hillslope frictions to vary up to 2 orders of magnitude greater than typical terrestrial reference values to allow for potentially unique supraglacial processes ranging from fast flow over smooth bare ice to slow porous-media flow through weathering crust. Finally, we bound $R_{\text{coef}}$ to range between 0.3 and 2.0 to allow for both the over- and underproduction of water excess without

imposing mass (e.g., runoff) production. For each of our 16 experimental trials, the EA thus solves for the optimal combination of hillslope and channel friction in tandem with runoff production to best match the ADCP record measured at the outlet. Recall we do not run the SMB models directly.

We parameterized our EA as follows. Crossover probability and distance were set to 0.7 and 5, respectively, and mutation probability and distance were set to 0.2 and 10, respectively. These parameters control the degree of change in one parameter set to the next. The objective function for the EA was the Nash–Sutcliffe efficiency (NSE) at the outlet, calculated between the in situ ADCP record and the model discharge. NSE is a standard hydrology metric for hydrograph analysis which is optimal at a value of 1. An NSE of 0 is equivalent to modeling a hydrograph as the true mean flow, and negative NSE values indicate that the mean outperforms a given model. Finally, we set the population size and number of generations (parameters that control how many different solutions the EA tests for CE7 the size of the search space in tandem with crossover and mutation) based on the model configuration (e.g., fine networks with hillslope processing take much longer to run and therefore used less generations; see below) due to runtime. Even though we ran our tests us-

**Table 1.** Field-based constraints on HRR routing model parameters (from the literature and Gleason et al., 2016).

| Parameter | Min | Max | Upstream area (km$^2$) |
|---|---|---|---|
| Hillslope friction | 0.05 | 25 | n/a `TS12` (global parameter) |
| $R_{coef}$ | 0.3 | 2.0 | n/a (global parameter) |
| $n_1$ | 0.0050 | 0.0600 | area $<0.010$ |
| $n_2$ | 0.0045 | 0.0600 | $0.010<$area$<0.025$ |
| $n_3$ | 0.0040 | 0.0600 | $0.025<$area$<0.063$ |
| $n_4$ | 0.0035 | 0.0600 | $0.063<$area$<0.200$ |
| $n_5$ | 0.0030 | 0.0600 | $0.200<$area$<0.500$ |
| $n_6$ | 0.0025 | 0.0600 | $0.500<$area$<1.260$ |
| $n_7$ | 0.0020 | 0.0600 | $1.260<$area$<3.160$ |
| $n_8$ | 0.0015 | 0.0600 | $3.160<$area$<10.000$ |
| $n_9$ | 0.0010 | 0.0600 | area$>10.000$ `TS13` |

ing parallel computing on a powerful modeling machine (Intel Xeon Gold 6126 3 GHZ CPU with 96 GB of RAM and 24 logical processors), a single fine-network hillslope HRR run took approximately 2 min to complete. Thus, we used a population size of 40 for the non-hillslope tests, 16 members for the coarse hillslope test, and 12 members for the fine hillslope test. EA length was set to 2500 generations for the non-hillslope tests and 1000 and 500 generations for the coarse and fine hillslope tests, respectively. The total number of tested parameterizations is equivalent to the number of generations multiplied by the population size, so we tested between 6000 and 100 000 parameter sets across our calibration runs, equivalent to approximately 6 d of computing time for the longest calibration. We saved globally optimal results as they occurred within the EA as a single objective problem, and these results were obtained well before the end of the EA in each run, so we are confident that the length of the EA was sufficient in each case.

## 4 Results

### 4.1 Basin outlet hydrograph

We first analyze our model results at the basin outlet (Fig. 3). In aggregate, two major results are immediately apparent across our 16 model configurations. First, the fine river network generally outperformed the coarse network across models and hillslope choices (as seven of the eight fine networks appear in the top 10 performing models; Table 2). Second, the top three performing models all include explicit hillslope kinematic wave routing, with the best outcome (a RACMO2-forced fine-network hillslope configuration) hav-

ing an excellent calibration RMSE of 1.85 m$^3$/s. Model calibration statistics show high skill (defined here as NSE $>0.8$) in 5 of the 16 cases and moderate skill (NSE $>0.5$) in all 16 cases, with RMSE ranging from 1.85 to 4.55 m$^3$/s (observed flows ranged from 4.6 to 26.7 m$^3$/s, for context). Note that RMSE and NSE do not track perfectly given the differing nature of their assessments. RMSE is a total mass error that is influenced by the scale of variation in the hydrograph, in which NSE compares to the mean. There is no universally acknowledged threshold for model calibration goodness of fit, but the models presented here meet a traditional gauging station expectation of 5 %–10 % error in matching ADCP flows (Turnipseed and Sauer, 2010).

All 16 calibrated HRR model configurations match daily peak flow magnitude and timing, regardless of input runoff or hillslope/density controls. This occurs despite runoff forcings from each model that are out of phase with the peak recession observed in the ADCP outlet hydrograph. While all calibrated models match peak magnitude well, only RACMO2-forced models capture the peak recession seen by the ADCP. All instantaneously routed SMB runoff incorrectly shows zero flow in the overnight period, and many of our calibrated models also approach near-zero flow overnight, but the fine-network models do correctly retain some water regardless of forcing. RACMO2-forced experiments are successful at matching both peak and low flows for all experiments except the coarse non-hillslope case and indeed achieve NSE scores of up to 0.92 and a corresponding RMSE of only 1.85 m$^3$/s. Post-routing total cumulative discharge is relatively consistent across all models (see Fig. 4 where total discharge is shown for hillslope models). $R_{coef}$ varied from model to model but little within each model and ranged from 41 % to 88 % retention (Table 2). Despite indicating that reduced input runoff is required to route flows accurately across all models, overall routed cumulative discharge was lower than in situ measurements for this time period for coarse networks due to the underprediction of low flows, and it was overpredicted using fine networks (Fig. 4).

Finally, we calculate routing delays for each of our 16 calibrated routing models by noting the difference in ADCP peak and the unrouted SMB runoff peak. Routing delay is a function of both time of day and discharge, but it is easiest to interpret at daily peak flow. This peak delay is the shortest for MERRA2 (1–3 h) and longest for MAR and RACMO2 (5–6 h). These values represent an estimate for daily peak flow delay between runoff forcing and the calibrated HRR model and represent the total travel time for water to pass through the system from runoff production to the outlet. Our routed flows are non-zero in many cases despite a zero water excess forcing at night (Fig. 3), signifying that the network architecture and HRR-modeled routing delays are sufficient to introduce physically realistic (i.e., non-zero) nighttime water discharges atop the GrIS, consistent with in situ ADCP measurements in Rio Behar.

**Table 2.** Calibrated parameters for all 16 coupled SMB–HRR model experiments. Table is ranked by NSE per row, with the top performing model in the first row.

| Experimental setup | | | Calibrated model parameters | | | | Performance metrics | | |
|---|---|---|---|---|---|---|---|---|---|
| SMB forcing | Hillslope | Network density | $R_{coef}$ | $\overline{n}$ | $n_{sd}$ TS14 | Hillslope friction | NSE | KGE | RMSE ($m^3$/s) |
| RACMO2 | Included | Coarse | 0.50 | 0.027 | 0.026 | 13.64 | 0.92 | 0.96 | 1.85 |
| RACMO2 | Included | Fine | 0.49 | 0.008 | 0.014 | 25.00 | 0.89 | 0.87 | 2.17 |
| MAR | Included | Coarse | 0.66 | 0.011 | 0.017 | 14.34 | 0.89 | 0.94 | 2.19 |
| RACMO2 | Excluded | Fine | 0.50 | 0.015 | 0.022 | – | 0.86 | 0.92 | 2.49 |
| MAR | Excluded | Fine | 0.63 | 0.019 | 0.025 | – | 0.80 | 0.83 | 2.96 |
| HIRHAM | Excluded | Fine | 0.47 | 0.026 | 0.019 | – | 0.79 | 0.77 | 3.03 |
| MERRA2 | Excluded | Fine | 0.84 | 0.021 | 0.024 | – | 0.76 | 0.71 | 3.20 |
| MERRA2 | Included | Coarse | 0.88 | 0.006 | 0.007 | 5.44 | 0.75 | 0.73 | 3.31 |
| MAR | Included | Fine | 0.61 | 0.016 | 0.019 | 0.05 | 0.74 | 0.75 | 3.35 |
| MERRA2 | Included | Fine | 0.82 | 0.016 | 0.019 | 0.05 | 0.71 | 0.75 | 3.51 |
| MERRA2 | Excluded | Coarse | 0.80 | 0.025 | 0.024 | – | 0.64 | 0.59 | 3.95 |
| HIRHAM | Included | Coarse | 0.48 | 0.006 | 0.007 | 1.78 | 0.62 | 0.63 | 4.03 |
| MAR | Excluded | Coarse | 0.55 | 0.044 | 0.025 | – | 0.60 | 0.57 | 4.15 |
| HIRHAM | Included | Fine | 0.47 | 0.031 | 0.021 | 0.05 | 0.57 | 0.60 | 4.30 |
| HIRHAM | Excluded | Coarse | 0.46 | 0.022 | 0.026 | – | 0.56 | 0.56 | 4.37 |
| RACMO2 | Excluded | Coarse | 0.41 | 0.055 | 0.012 | – | 0.52 | 0.59 | 4.55 |

## 4.2 Lower-order hydrographs

While we cannot verify flows at any network channel besides the outlet, we have simulated hourly flows for all 1044 and 8095 channel segments in the coarse and fine networks, respectively. If we assume that accurate model performance at the main basin outlet indicates physically realistic upstream flows, it is profitable to report results for upstream flows during the calibration period. To analyze these large datasets, we summarize flows in the 72 h validation period by stream order, with Fig. 5 presenting results for 1st–3rd order streams and Fig. 6 presenting results for 4th and 5th order streams. In each figure, we plot the mean hydrograph for the order with 1-standard-deviation shaded area to represent variability around the mean. Geomorphic theory predicts a geometric decline in the number of streams per order (Allen et al., 2018), and thus orders with fewer streams are more homogenous by definition in these plots.

There is a large difference in flow magnitude across fine and coarse models regardless of SMB forcing or inclusion/exclusion of hillslopes (Figs. 5, 6). For 4th and 5th order streams these flow differences span roughly a factor of 2, while in the lower orders flow differences span almost an order of magnitude. This signifies that smaller streams are more sensitive to their hillslopes, as expected. We also note that the networks have different total orders (six for the coarse network, seven for the fine network). Therefore, the 2nd order fine streams loosely correspond to 1st order coarse streams, but this correlation is not a 1 : 1 match. Peak timing also differs between hillslope and non-hillslope models in the lower orders for coarse networks. This effect is more pronounced in the lowest 1st–3rd orders, in which, e.g., RACMO2-forced models show a peak delay of almost 5 h between hillslope and non-hillslope models. This delay in peak timing when explicitly modeling a hillslope process at smaller streams is intuitive and stronger in coarse models, which have larger individual hillslopes via their larger channel inception area threshold.

Turning to the calibrated model parameters, mean $n$ values (across either 1044 or 8095 channels) ranged from 0.006 to 0.055 across all 16 calibrated models (Table 2). The standard deviation of $n$ varied considerably and was often the same order of magnitude as its mean (Table 2). Figure 7 summarizes channel friction across the inclusion/exclusion of hillslope (e.g., hillslope/non-hillslope) process and across coarse/fine networks. Channel friction is given by calibrated $n$, and recall that we calibrated channel friction in nine discreet bins based on upstream area such that all channels within the area bin receive the same $n$. Upstream area loosely tracks stream order, and thus the larger the area, the higher the order.

Non-hillslope large channels in the three highest orders require a substantially larger Manning's $n$ value than these same channels with a hillslope process included, indicating that the non-hillslope models necessitate higher friction in large channels to match outlet flows. For the second and third largest bins, this resulted in extreme friction in those channels just before the basin outlet in order to provide enough friction to conserve mass and momentum. For the lower order streams with upstream areas less than TS15 1.260 km², channel friction decreases with increasing upstream area. This pattern repeats when analyzing across coarse/fine networks, but there are less clear patterns in $n$ when analyzing the

**The Cryosphere, 15, 1–18, 2021** https://doi.org/10.5194/tc-15-1-2021

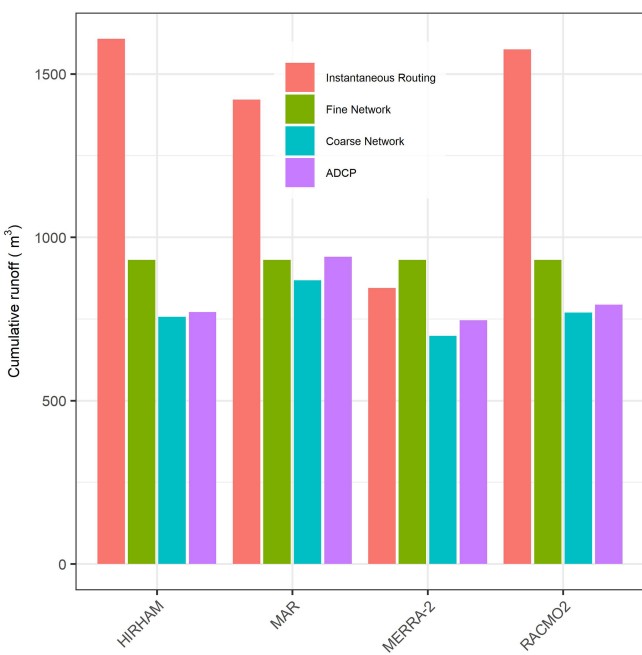

**Figure 4.** Total cumulative discharge for hillslope-enabled scenarios for the 72 h ADCP measurement period. Total water export is relatively consistent across all four SMB models but substantially different than input runoff (i.e., instantaneous routing) for all models but MERRA2. The ADCP represents a measured cumulative export, while instantaneous routing assumes that SMB runoff immediately leaves the watershed as soon as it is produced. Calibrated models underpredict water export due to the underestimation of nighttime low flows for coarse networks and overpredict total water export with fine networks.

coarse vs. fine network for the three largest bins. This suggests that the dominant control on modeled channel friction is whether or not water first enters a channel via a hillslope. Finally, channel friction values in Fig. 7 fall well within our physically realistic constraints until the three largest bins. These largest channels for non-hillslope models in particular require friction near the upper limit of plausibility (particularly the second largest bin) to satisfactorily conserve mass, and the worse validation metrics for these configurations might be traced to this effect.

## 5 Discussion

We have successfully calibrated a hillslope river routing model capable of simulating hourly flows through thousands of supraglacial channels atop the GrIS while conserving runoff mass and momentum. The most accurate models to emerge from our experiments were those that employed a fine-channel network and/or inclusion of hillslope flow routing. We assert that our results support the inclusion of realistically fine river and/or stream networks and hillslope-enabled routing models for supraglacial runoff modeling ap-

plications that require the realistic representation of runoff timing and magnitude. While we cannot validate in-channel flows upstream of the outlet, this level of hydrological simulation could, for instance, be coupled with SMB models to calculate hourly moulin discharge rates, lake fill-and-spill volumes, channel incision rates (e.g., following Karlstrom and Yang, 2016, or Koziol et al., 2017), and supraglacial contributions to subglacial water pressures (e.g., following Banwell et al., 2016 or Yang et al., 2020). These processes have important implications for GrIS surface hydrology, surface mass balance, and subglacial hydrological systems. We believe this work represents a promising step toward coupled SMB-routing modeling that can be used to generate more realistic predictions of these processes and their sensitivity to changing surface meltwater forcings or surface topography.

The goal of this study was not to interrogate individual SMB models or suggest one is better than another. A recent synthesis (Fettweis et al., 2020) showed that SMB models vary considerably given the same forcing, and readers are referred to this and other literature for further information on why these models might disagree. Smith et al. (2017) and Mankoff et al. (2020) have both explored what these differences mean for water exiting the GrIS, but our purpose is to demonstrate the importance of coupling SMB model output with a surface flow routing model to understand runoff transport before it enters the englacial system. This enables rigorous estimation of supraglacial flow accumulation and routing delays to moulins atop the GrIS that route meltwater into a dynamically varying subglacial hydraulic system that influences ice sheet acceleration in response to the timing and magnitude of input discharges, which is imperative to accurately estimate diurnally varying moulin discharges using climate models. Second, this work advances the physical understanding of ice sheet surface hydraulic properties, for example, our finding hillslope friction values (Table 2) well outside typical terrestrial values of 0.01 to 1 (Hergarten and Neugebauer, 1997; McCuen, 1998 TS16; Kalyanapu et al., 2009). Yang et al. (2018) similarly estimated slow transport of meltwater on ice interfluves (similar to the hillslopes studied here) some 2–3 orders of magnitude slower than open-channel flow ($\sim 10^{-1}$ m/s TS17). Observations of ice density and saturation in shallow ice cores within the Rio Behar catchment indicate that substantial subsurface meltwater is stored within the upper decimeters of bare-ice weathering crust and was anecdotally observed to percolate through the crust (Cooper et al., 2018). If so, this unsaturated flow would move orders of magnitude slower than bare-ice overland flow. These convergent findings are consistent with conceptual models of unsaturated subsurface porous media flow and support the very slow lateral transport we observe here (on the order of $10^{-5}$ to $10^{-1}$ m/s) to the channel from the ice surface, but we cannot make any further conclusions on physical processes or mechanisms given our experiment design and model setup. That is, since we lump all flow over and through the ice, firn, snow, and crust before it reaches

https://doi.org/10.5194/tc-15-1-2021

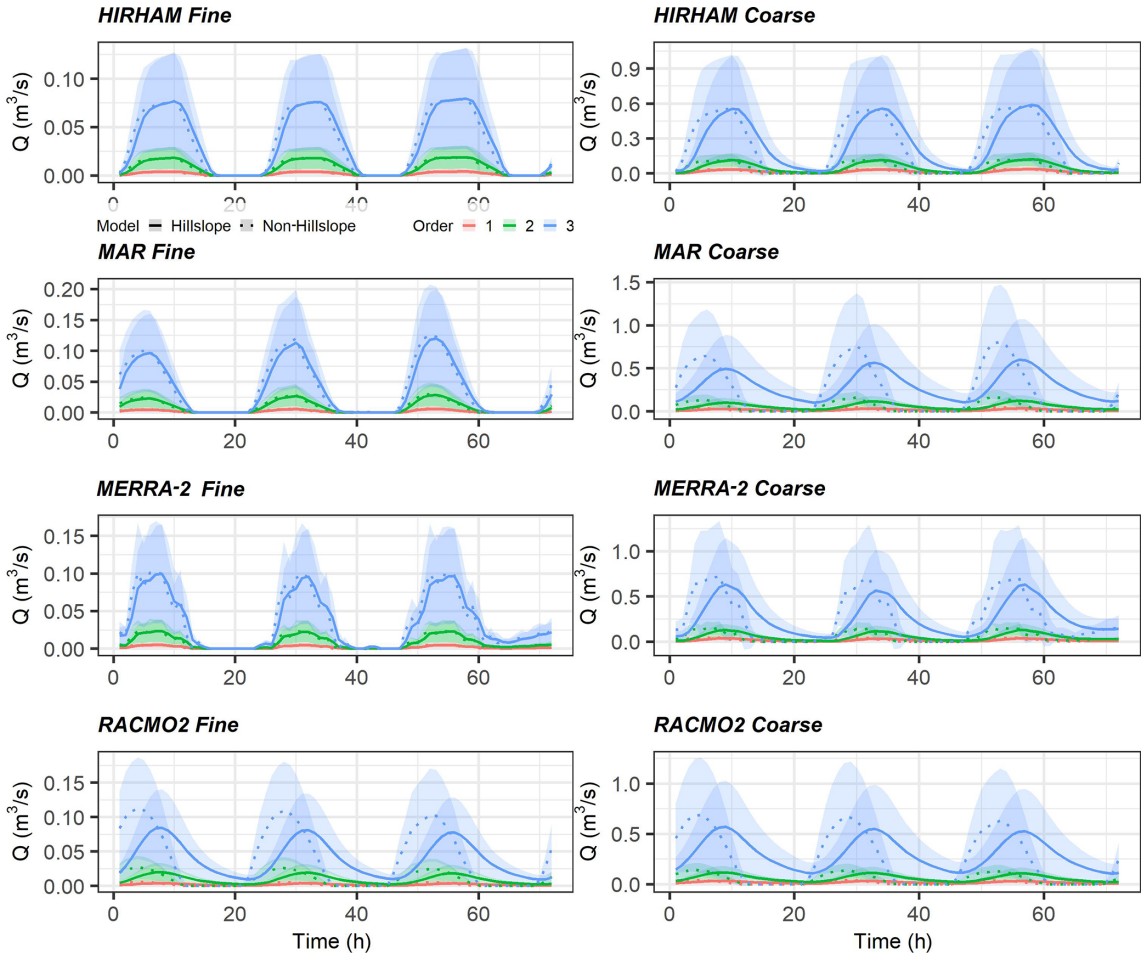

**Figure 5.** Mean and 1-sigma shaded variability for channel segment hydrographs by order for 1st–3rd order streams for the validation period. Non-hillslope process flows are dashed. Note the increase by a factor of ∼ 10 in flows between fine and coarse networks and the difference in peak timing between hillslope and non-hillslope models.

channels into a single "hillslope" flow with a single friction, we can be confident in the speed of this transport but not its flowpaths or mechanism. This result highlights the need for further basic research on the supraglacial hydrological pro-
5 cess to further understand the importance of these velocities.

The importance of including hillslope process is also clearly manifested through calibrated channel frictions generated in model experiments that exclude it. There are discernible changes in channel friction when hillslopes are or
10 are not modeled, and the results are intuitive: channels lacking hillslopes have much higher friction, especially in large channels (Fig. 7). Further, for the largest channels (i.e., upstream areas greater than $1.260\,\mathrm{km^2}$ CE9), models without hillslopes take channel friction values almost uniformly at
15 the maximum of the realistic constraints we set (Fig. 7) while at the same time having a poor match to observed flows (Table 2, Fig. 3). HRR is not a glaciological model, and therefore it is agnostic CE10 about sources of friction and can trade off channel and hillslope friction to produce correct outflows if

unconstrained. We have constrained the channel friction to 20 match literature field observations closely and allowed hillslope frictions to vary over a much wider range of values given the longer history of study and larger databases of Manning's $n$ values for ice channels relative to transport through the crust and/or bare ice. Therefore, non-hillslope models would 25 likely improve only by including physically unrealistic channel frictional values given the results in Fig. 7. This is in line with mass conservation, as without hillslopes to slow water upstream, HRR needs to slow water using extreme friction near the outlet in order to match the hydrograph. This pattern 30 is observed across both coarse and fine networks.

Ideally, we would have enough data to calibrate and validate the model over separate time periods and at more locations than the outlet. HRR produces an individual hourly discharge at each of our thousands of channels, but we can only 35 verify these at the outlet. However, we believe that model calibration statistics at the outlet indicate the physical realism of the process we are attempting to model: since we

https://doi.org/10.5194/tc-15-1-2021

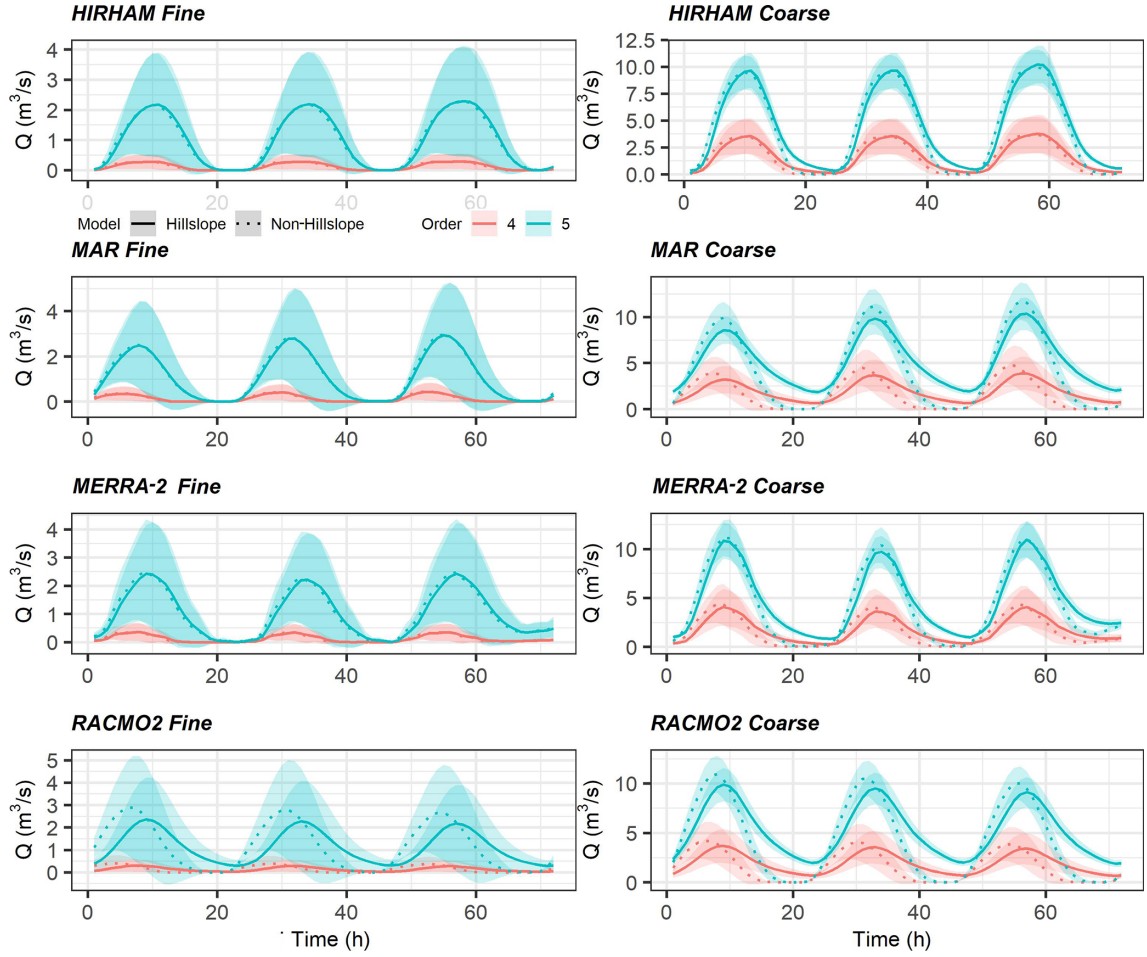

**Figure 6.** As Fig. 5 but for 4th and 5th orders. Note the increase by a factor of ∼ 2 in flows between fine and coarse networks and the reduction in variability in coarse network flows. As before, the shaded areas represent variability, not uncertainty.

modeled an accurate outlet hydrograph, the fully mass- and momentum-conserved physics of HRR mean that upstream flows must be realistically represented or we could not have produced a quality outlet hydrograph. Our results show that HRR is capable of matching outlet flows extremely well (calibration Kling–Gupta efficiency, KGE, as high as 0.96 and NSE as high as 0.92), and thus we believe this assumption well-founded. Recall also that the ADCP data were collected from July 20 to 23, but we model hourly flows for the entire month. We focus our evaluation only on this 72 h calibration period to discuss our experimental results without discussing the rest of the month's unverified results. Results for these other times are of course an ultimate end goal of future GrIS water routing as we look toward future coupled SMB-routing models that can be used to study interactions between surface hydrologic routing processes and subglacial processes. While we have here only reported flows during a verifiable 72 h period, in theory our model parameters should be able to accurately route water in similar areas of the GrIS with similar network drainage patterns in similar seasons.

Our results also support earlier assertions of the mismatched timing and magnitude of SMB runoff and observed discharges entering the Rio Behar terminal moulin (Smith et al., 2017). The routing model is unable to assign glaciologic process to mass gaps, so we can only suggest plausible mechanisms for closing that mass balance gap. Mass gaps could perhaps result from subsurface retention and/or refreezing in bare-ice weathering crust (Cooper et al., 2018), a process not currently well-represented in SMB models, or the mass imbalance could come from transport processes: filling lakes, drainage through fractures (there are no crevasses in the study area), or the breach of topographic divides are all plausible transport process gaps. Topographic breach is unlikely given that we use an observed (via image) channel network, and thus if breaches did occur, they are accounted for. Further, total depression storage (including true lakes and DEM artifacts) was $6.92e^6$ m$^3$ TS18, which is 2 orders of magnitude less than the observed ADCP flux during this time (integrated into a bulk volume, $241e^6$ m$^3$) and 1 order of magnitude less than the maximum runoff deficit (obtained by sub-

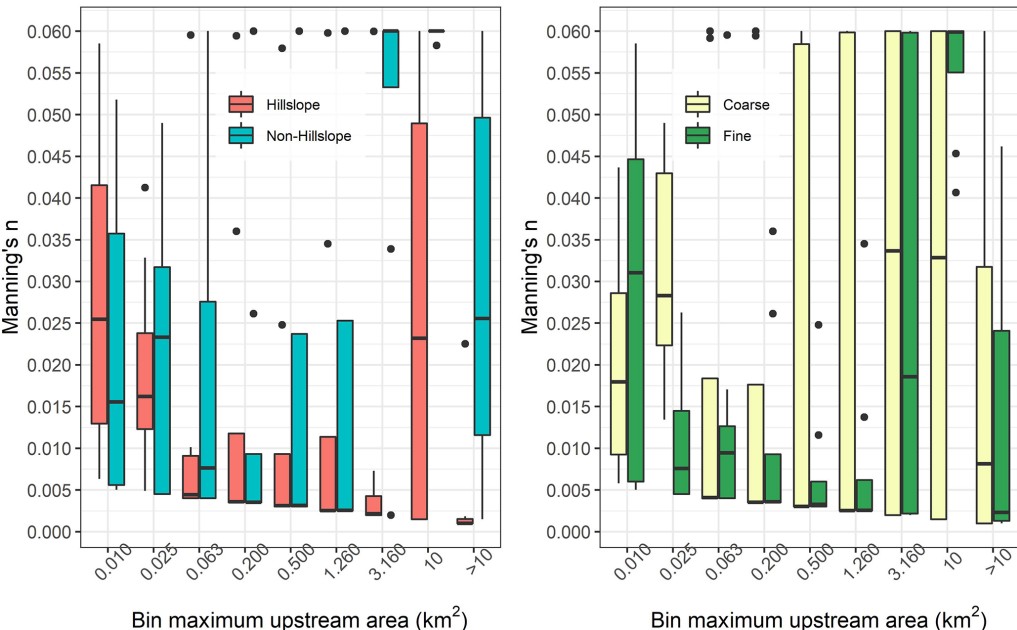

**Figure 7.** Mean Manning's *n* for all rivers binned by area, in which bin refers to an area threshold given in Table 1. Bins are bounded by the maximum value indicated on the *x* axes and a minimum value equal to the maximum area of the next smallest bin. There are eight values per each boxplot: these represent the mean Manning's *n* for all channels in that area bin for each of eight experimental trial configurations. Our experiment design yields, for instance, eight models that include hillslopes (four of which are coarse, and four of which are fine), and these boxplots plot the mean *n*, per bin, of those eight models. Boxplots are standard and show median, interquartile range (IQR) CE8, and outliers. Non-hillslope trials require substantially more friction than hillslope trials in the largest channels, suggesting compensation for lack of hillslope process representation.

tracting the ADCP from the largest SMB input, $17.5e^6$ m$^3$). Therefore, if all depressions were dry at the start of routing and were completely filled by runoff before beginning to flow in the channel network, this would still only account for 5 roughly one-third of extra runoff production mass. Given that we know lakes are full during this time period, we assert that this lake filling effect is not the cause of mass imbalance. Further, errors in our outlet hydrographs are dominated by the underestimation of nighttime low-flow periods as peak flows 10 are modeled well across nearly all 16 trials. These nighttime low flows are particularly important for mass balance in the Rio Behar watershed as a large driver of mismatches in total mass balance (Fig. 4) comes from these low-flow periods. Error could come from the ADCP itself, and this instrument 15 is generally less certain at lower flows. However, the ADCP record here is taken from Smith et al. (2017) and represents a well-documented procedure carried out by expert field personnel, and thus we are confident that ADCP errors are too small to explain $R_{coef}$. We affirm that all SMB models examined here produce too much excess water relative to ADCP observations (at least at peak times, Fig. 4 shows MERRA2 total runoff is less than the ADCP total discharge but still requires $R_{coef} < 1$ to reduce the peak daytime volume of water) and do not model nighttime flows without routing, consistent 25 with Smith et al. (2017). Our results suggest that hydrologic

process modeling (i.e., routing) can correctly reproduce these nighttime low flows.

The workflow presented here is repeatable for any supraglacial stream and river network on the GrIS, but the in situ discharge datasets needed for calibration are not readily available. Future studies attempting to repeat this model setup elsewhere need an in situ discharge record (ideally longer than our 3 d record and ideally collected at multiple locations across stream orders), a high-quality DEM, and a fine-scale remotely sensed image. Modeling is efficient with these data in hand, yet the collection of in situ discharge in particular presents a major hurdle for widespread application to the GrIS. It is possible to use assumed discharges for calibration, but as our results clearly support a difference between predicted and measured fluxes, we believe measured calibration data are best. We suggest that the collection and publication of a repository of supraglacial channel discharges and hydraulic properties atop the GrIS would be an invaluable resource and that future studies should explore the transferability of key parameters (e.g., channel and hillslope frictions) to other locations on the ice sheet.

# 6 Conclusions

We confirm earlier assertions of the importance of terrestrial hydrological processes, specifically hillslope water transport and open-channel flow, on GrIS surface meltwater routing. Unlike previous studies routing meltwater, our results are generated using the hillslope river routing model (HRR) which uses an explicit kinematic wave to conserve water mass and momentum in hillslopes and channels and represents hourly flow in nearly 10 000 individual channels in a fully topological network. This first-principles investigation shows that observed supraglacial river discharges (and thus moulin hydrographs) cannot be accurately simulated without both reducing the volume of surface runoff generated by SMB models and accounting for hydrologic transport processes. We investigated two process-level controls on this modeling – modeling coarse- vs. fine-scale channel networks and inclusion/exclusion of hillslope process – and found that incorporating fine-scale channel networks and hillslopes yields superior results. Calibrated model parameters are intuitive and align with field observations and theory. The automated methods developed here could readily be deployed elsewhere atop the GrIS bare-ice ablation zone but require in situ supraglacial discharge data for calibration. More of these data should be collected if GrIS surface hydrology processes are to be fully understood.

*Code and data availability.* TS19 All data and models used in this study were previously published and are accessible via their original publications as cited in the text.

*Author contributions.* CJG and KY conceived of the idea and designed the study. KY and KL extracted river networks. CJG and DF set up and calibrated HRR. CJG designed and created the figures and drafted the text. All other authors participated in fieldwork to collect the ADCP record, and all authors wrote the text. CE11

*Competing interests.* The authors declare that they have no conflict of interest.

*Acknowledgements.* We thank Ed Beighley of Northeastern University for developing and sharing HRR source code with us. Kang Yang acknowledges support from the National Key R&D Program (2018YFC1406101), the National Natural Science Foundation of China (41871327), and the Fundamental Research Funds for the Central Universities (14380070).

*Financial support.* This research has been supported by the National Key R&D Program (grant no. 2018YFC1406101), the National Natural Science Foundation of China (grant no. 41871327), and the Fundamental Research Funds for Central Universities of the Central South University (grant no. 14380070) TS20.

*Review statement.* This paper was edited by Nanna Bjørnholt Karlsson and reviewed by Sammie Buzzard and Ian Willis.

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

## Remarks from the language copy-editor

## Remarks from the typesetter

TS31  Please provide publisher and publisher location in case of a book. In case of a journal, please provide journal title and page range or journal title, article number and DOI.

TS32  Please provide page range or article number.

TS33  Please provide page range or article number.

TS34  Updated to the final version. The year was changed to 2020 in the text.