# Peer review of "Hourly surface meltwater routing for a Greenlandic supraglacial catchment across hillslopes and through a dense topological channel network"

_The Cryosphere, 2020_

## Referee Comment (RC1) · Anonymous Referee #1 · 8 Dec 2020

This paper details a study using the Hillslope River Routing Model (HRR) to better constrain surface meltwater routing on Greenland, improving on results that only use surface mass balance models. This is a good example of taking more advance techniques used in hydrology and applying them to glaciology, something which can greatly benefit the field.

However, I don't find the way in which the paper is currently written to be suitable for the target audience of a journal such as The Cryosphere. Many of the terms and methods which may be more familiar to trained hydrologists will not be familiar to many glaciologists. I would very much like to see this paper published in The Cryosphere

as it would be a good step towards better collaboration between these fields, but I would recommend a detailed re-write to make it more accessible to those without a background in hydrology.

My detailed comments are below:

Line 34: and lakes?

Line 40: Also Leeson et al. 2012 DOI: 10.5194/tc-6-1077-201

Line 82: What about subglacial channels/ transport through firn?

Line 97: Is this the only data available for the whole of Greenland? It seems a big weakness of this paper is the lack of validation data available. If other datasets were not used it should be explained why (but if this is really all there is then fair enough!).

Line 122: Related to the previous comment, is there data for other times available? Can you explain why it is so important only the peak of the melt-season can be used?

Line 129-133: Were other images available? What if other streams are present and just didn't have water flowing on this particular day?

Line 134: Please define DEM

Line 142: How do you determine all of the water in each of these grid cells will remain in this catchment, could any of it be being transported elsewhere?

Line 167: Please define Froude number.

Line 163 onwards: Why was none of this data suitable to use to validate the model?

Line 176: I know what you're getting at here but I'm not sure 'factorial' is the correct description?

Line 185: Is it possible that the SMB values are correct and water just isn't making it to the channels where the ADCP measurements are made? E.g. What about refreezing in firn?

Figure 1: The middle 4 squares could be laid out in a clearer way. Why does including or excluding hillslope flow have a 'yes' or 'no' but there is no differentiation between coarse and fine network?

Line 195-6: 'generalizing the process of water routing from satellite image collection to water routing' doesn't make sense to me.

Line 195 onwards: This paragraph was one I especially felt could be better explained for the non-hydrologist. It would also be good to see justification for 'standard hydrologic' practices being applied to icy surfaces.

Line 210-11: 0.5 is more than twice 0.2 so I wouldn't say they are matching, does this difference matter?

Line 217 onwards: Again this paragraph could be clearer. The authors discuss stream order again in the results so an introduction to what this means and why this matters would be helpful.

Line 234: Do any other models have this rigorous representation? If they are written in other, more accessible, languages it may still be useful to briefly mention them for those who may want to build on this but not use FORTRAN.

Line 245: Please give a little more detail about how EAs work.

Line 255: Please define Manning's n.

Line 296-7: Please define acronyms.

Line 311 and figure 4: Please comment more on why we are assuming here that MERRA2 is incorrect. Could it not be that all the other inputs are less accurate and MERRA2 is actually getting it right?

Line 402 onwards: How is the slow lateral transport accounted for in the model?

Line 420: Can you give an example(s) of the physical processes that may be leading

to this to support this conclusion?

Line 443: Anywhere or just in the bare-ice ablation zone?

---

## Referee Comment (RC2) · Ian Willis (Referee) · 13 Jan 2021

This is a novel piece of research addressing an important area of glaciology. It uses a well-established 'off the shelf' hillslope-channel hydrological model used previously in terrestrial settings, to route meltwater across a supraglacial catchment on the Greenland Ice Sheet. It provides a useful steppingstone, therefore, to one day developing fully coupled surface mass balance – surface water routing – subglacial water routing – glacier dynamics models. The paper adds to valuable previous work produced by this group. A series of experiments are created using four different runoff series from widely cited surface mass balance models (HIRHAM5, MAR3.6, RACMO2.3, and MERRA-2)

[Figure]

and for routing model sets ups that consider a course and a fine density channel network, and which either consider or do not consider hillslope routing processes. So 16 separate experiments are performed in total. Model parameters are calibrated by comparing outputs with measured discharges in a surface stream at the catchment outlet over a few days in July, which have been published elsewhere. Parameter space is explored, and the patterns of parameter values are used to infer the importance of catchment processes, for example the role of hillslope friction, which is high compared to terrestrial settings and which, it is argued, may represent shallow subsurface routing through a weathering crust.

The paper is nicely structured and generally well written, and the Figures and Tables are clear and useful. The work is thorough, generally acknowledges previous work (with a few exceptions that could be added at the authors' discretion – see below) and provides a valuable contribution to the literature.

There are five places where I think things need to be explained in more detail or where the results could be discussed further. These are:

1. On line 210/11 you tell us the channel widths that are produced for the fine network. But how are channel widths determined? Was this explained? I'd assumed they'd be dictated by the DEM grid size? Why don't you also tell us the widths of the channels produced for the coarse network here?

2. On line 222. You say "and lakes are represented by wide, shallow 'throughflow' river segments…" Is this a major limitation of this work? How do lakes fill and drain? Surely, the filling and draining of lakes will have a major impact on the relation between distributed runoff and the hydrographs at the catchment outflow and yet this important process is not incorporated. I'd like to see more discussion of this. How many sinks are there in the catchment that need to be filled? Where are they? What are their volumes? What are the implications for water routing?

3. On line 397 you show that after calibration, hillslope friction values were on occasion

very high – up to the max. threshold allowed of 25. These do seem very high. To what extent are these high because you didn't allow channel frictions to rise very high (Table 2)? Could you explain a bit more about the range of parameter values considered in Table 2 and the implications of this?

4. Related to point 2 above. On lines 398-405, you find very slow water transport rates on hillslopes here and suggest this may be due to slow transport through an ice crust. But what role does ignoring travel through lakes as they fill up and overtop their outlet channels play on your results?

5. You need to introduce a runoff correction coefficient (Rcoef) to match modelled with measured hydrographs and this turns out in the calibration to always be less than one which means runoff from all the SMB models is over predicted or measured discharge at the catchment outlet is under predicted. This is discussed to some extent in the Discussion but I think more thought could be given to this. You suggest water may be stored (and possibly freeze) in a weathering crust but is this feasible? What volumes are we talking about and could storage in a weathering crust really explain it? Related to above, is it possible lakes may be filling over this period which would explain the discrepancy. Could it also be due to leakage into the ice sheet via crevasses?

Other than these five points, I have just a series of questions / suggestions regarding improving clarity as follows.

L27. 'routed to match measured flows' could be deleted as it's repetition.

L29/30. '. . .explicitly including hillslope flow and routing runoff through a realistically fine channel network. . .' It's difficult to fully understand this without reading the rest of the paper. Could you somehow add "as opposed to not including hillslope processes" and "as opposed to a course channel network"?

L38. 'with unique and complex hydrologic process distinct from terrestrial hydrology' This is rather convoluted. Could it just be changed to 'which is distinct from those in

terrestrial settings'?

L48. Should 'ice' be changed to "glacier" or "ice sheet" or "snow, firn and ice"?

L54. After 'lake impoundment' you could consider referring to: Arnold, N.S., Banwell, A.F. and Willis, I.C., 2014. High-resolution modelling of the seasonal evolution of surface water storage on the Greenland Ice Sheet. The Cryosphere.

L63/4. As well as Banwell et al 2013, could also refer here to Banwell, A., Hewitt, I., Willis, I. and Arnold, N., 2016. Moulin density controls drainage development beneath the Greenland ice sheet. Journal of Geophysical Research Earth Surface, v. 121, p.2248-2269.

L64. Immediately before the sentence beginning 'Liston and Mernild (2012)' you could add another sentence summarising and referring to the work of Leeson et al 2012: Leeson, AA, Shepherd, A, Palmer, S, Sundal, A & Fettweis, X 2012, 'Simulating the growth of supraglacial lakes at the western margin of the Greenland ice sheet', Cryosphere, vol. 6, no. 5, pp. 1077-1086.

L75-77. Would it be better to say: "These previous efforts successfully modelled meltwater transport on the GrIS ablation zone, but their relative simplicity allows space for more sophisticated routing models from terrestrial hydrology to be applied to ice sheet surfaces"? Or something like this.

L80. '…at the global…'

L81. Is 'paradigm' a little grand? Would "approach" be better instead?

L85. What is meant by "explicit routing"? As opposed to implicit routing? What about saying "...accurately routing water at the catchment scale across the GrIS"?

L86. What is meant by "explicit" here?

L87/88. "…but to our knowledge no automated, large scale network extraction and topological connection work exists for the GrIS". Do you mean for the whole of the GrIS

here ? There are examples of this at smaller scale surely.

L89. "these issues" What issues? You've not explicitly referred to 'issues' above.

L90. "...a model such as..."

L90 change "science" to "understanding" ?

L92. What are "network scales"? You've referred to catchments and (I think) the entire ice sheet previously which I understand but what are these?

L93. Delete the word "these" as you've not referred to processes previously.

L93. Suggest change 'are interacting" to "interact".

L96. What is meant by "spatially explicit"?

L97/8. I think the semicolons would be better as periods.

L104. Would it be clearer to say "...test how the representation of hillslope processes and network density (as derived by our automated network generation process)..." or something like that?

L115. Delete "(Section 2.2) as you're referring to Smith et al 2017 here.

L116-119. Would it be clearer to say: "Previous work in the basin includes: i) a comparison of SMB runoff and field measured discharge using a simple routing method (Smith et al., 2017); ii) a study of subsurface water storage in bare-ice weathering crust (Cooper et al., 2018); iii) albedo mapping (Ryan et al., 2017); and iv) mapping the catchment's supraglacial channel network via satellite and un-crewed aerial vehicle (UAV) remote sensing (Ryan et al., 2017; Yang et al., 2018)"?

L120. Suggest change "...basin: we here..." to "...basin. Here we..."

L122. 'of a summer"

L122. Delete "as"

L124. Consider changing "...catchment. However..." to "...catchment, but the..."
L128/9. Perhaps add Banwell et al 2016 to this list?

L134. Would "...2 m resolution portion of the ArcticDEM was obtained..." sound better than "...2 m resolution ArcticDEM DEM was obtained ..."?

L134 "The ArcticDEM has been..."

L135. Could reference Pope et al, and Moussavi et al somewhere here as they use the ArcticDEM in the context of representing GrIS surface lake bathymetries

Pope, A., Scambos, T. A., Moussavi, M., Tedesco, M., Willis, M., Shean, D., and Grigsby, S.: Estimating supraglacial lake depth in West Greenland using Landsat 8 and comparison with other multispectral methods, The Cryosphere, 10, 15–27, https://doi.org/10.5194/tc-10-15-2016, 2016.

Moussavi, M. S., Abdalati, W., Pope, A., Scambos, T., Tedesco, M., MacFerrin, M., and Grigsby, S.: Derivation and validation of supraglacial lake volumes on the Greenland Ice Sheet from high-resolution satellite imagery, Remote Sens. Environ., 183, 294–303, https://doi.org/10.1016/j.rse.2016.05.024, 2016.

L138-143. Can you better explain here how you deal with the very different grid sizes between the 2m DEM and the various (I assume) grid sizes of the 4 models? How do you interpolate across the finer scale grid? Is the Rio Behar catchment entirely within a SMB model grid?

L154. Suggest "...hydrology and its discharge values are frequently..."

L154. Suggest "...(Gleason and Durand, 2020). Further reading..."

L157. Suggest changing "and ending" to "to".

L161-2. "allowing us to calibrate the free parameters of the routing model (Section 3.2) and to adjust water excess of the SMB models to best match these observations." Suggest delete this as this was explained earlier.

L174/5. I don't quite follow this sentence. Do you mean "Our overall goal for this study is to improve current understanding of supraglacial hydrological transport processes through application of a hydrological model, which includes classical hillslope and channel routing processes, to a catchment on the GrIS" ?

L192. "framework to guide"

L193. " Figure 1 shows an overall schematic of our approach." This sentence comes too late as Fig 1 has already been referred to. Could just delete or move to after first sentence of this section on line 175.

L195/6 "we are interested in generalizing the process of water routing from satellite image collection to water routing" doesn't make sense to me.

L199-203. I don't quite follow these two sentences. Could they say "Having done this, conventional network generation was confounded due to two large topographic depressions, one located in the upper part of the catchment and one located near the catchment outlet. Standard DEM preparation for hydrological analysis (in which the upstream depression was filled while the outlet depression was preserved) generated unrealistic parallel drainage channels upstream and no channels in the outlet depression of the catchment" ?

L204 delete "Rio Behar"

L25 "ultimately"

L210/11 Why don't you also tell us the widths of the channels produced for the course network here? This would be useful. How are channel widths determined? I assumed they'd be dictated by the DEM grid size.

L212-214. Would it be better to say: "It would be possible to derive expected rates of incision (and additional meltwater supply) due to frictional heating of the channels, but without including a radiation budget and ice property data we could not model how…"

L215/16. I don't follow this sentence. Should this say " ...we model the networks as static snapshots within HRR, which is..." ? I assume networks should be plural as there is the coarse one and the fine one. Need to explain what you mean by "loosely coupled to SMB runoff" or delete this.

L217. Could delete "ultimately"

L219. Suggest "...higher order) and the fine network had seven..."

L222. You say "and lakes are represented by wide, shallow 'throughflow' river segments..." But where are these in Fig 2? Is this a major limitation of this work? How do lakes fill and drain? See also my main point 2.

L229/30. Unless I missed it, Section 2.1 does not describe how channel widths were derived. It refers to Section 3.2 but that doesn't describe it either. How are channel widths derived?

L232. "...Section 3.2 is required..."

L233-5. Sentence here seems misplaced. Better to justify using HRR earlier, when you first introduce it - i.e. Section 1.

L278-80. This sentence doesn't quite makes sense. Suggest rewrite to clarify.

L282. What is a "population member"? This is not explained.

L282/3. On L175 you refer to "two experimental settings (inclusion/exclusion of hillslope flow, coarse/fine channel network densities)". See also nomenclature used in Fig 1 e.g. "hillslope coarse" or "non hillslope fine". Can you use consistent terminology so it's clear that you're running tests that both include and exclude hillslope processes and also that the 'coarse' and 'fine' refer to the network density? The terms you use here "non-hillslope tests", "coarse hillslope test" and "fine hillslope test" are a bit sloppy and rather confusing and don't match earlier. See also legend in Fig 3.

L315-18. Confusion over whether you're defining routing delay to be generic or specifically in terms of comparison of peaks. Would it be better to say something like: "Finally, we calculate routing delays for each of our 16 calibrated routing models. Routing delay will be a function of both time of day and discharge but here we calculate routing delay as the difference in ADCP peak and the unrouted SMB runoff peak. This delay is the shortest for MERRA2 (1-3 hours) and longest for MAR and RACMO2 (5-6 hours). These values provide an estimate for daily peak flow delay between runoff forcing and the calibrated HRR model."

L349. "….large data sets…"

L350. "Figure 5"

L366/7. You say "across inclusion/exclusion of hillslope process and across coarse/fine networks". So this nomenclature matches initial terminology (line 175) but not that used in other places – see comment re L282/3

Fig 7 Heading. You say "Mean" but these are box whisker plots so explain what all features represent. What are the dots?

In the body of the paper, can you explain the reason for the very anomalous results for bin 3.16 km and 10 km where Manning's n is very high?

L382. Should 'and/or' just be 'and'? The runs that included hillslope processes and had a fine channel network were best right?

L387. In addition to ref. Karlstrom and Yang, 2016 you could also include:

Koziol, C., Arnold, N., Pope, A. and Colgan, W., 2017. Quantifying supraglacial meltwater pathways in the Paakitsoq region, West Greenland. Journal of Glaciology,

as they too model channel incision.

In addition to Yang et al, 2020 you could also refer to:

Banwell, A., Hewitt, I., Willis, I. and Arnold, N., 2016. Moulin density controls drainage

development beneath the Greenland ice sheet. Journal of Geophysical Research Earth Surface, v. 121, p.2248-2269.

and

Koziol, C.P. and Arnold, N., 2018. Modelling seasonal meltwater forcing of the velocity of land-terminating margins of the Greenland Ice Sheet. The Cryosphere, v. 12, p.971-991

as they too couple SMB models to surface and subglacial routing models to examine basal water pressures.

L398-405. You find very slow water transport rates on hillslopes here and suggest this may be due to slow transport through an ice crust. But what role does ignoring travel through lakes esp. while lakes may be filling play on your results?

L427. You say "...parameters should be able to accurately appliable to flow route watering in..." This is not grammatically correct. Also, what is 'watering'? I'm not sure this statement is correct is it? Don't you imagine there is a huge evolution on some of these parameter values over the summer?

L434-5. Regarding this sentence, lakes may be filling over this period which would explain the discrepancy. Could it also be due to leakage into the ice sheet via crevasses?

---

## Author Comment (AC1) · 10 Feb 2021

*Below, find the original reviewer concern/question in regular typeface followed by our response in italic typeface. This response is written in anticipation of making all corrections to a manuscript as stated. Therefore, we make statements such as 'we changed' or 'we corrected' or 'we modified' in past-tense as if this document were being submitted together with a revised manuscript. It is our intent that the revised manuscript will read exactly as suggested here. We have also included references to the other review.*

Reviewer:
This paper details a study using the Hillslope River Routing Model (HRR) to better constrain surface meltwater routing on Greenland, improving on results that only use surface mass balance models. This is a good example of taking more advance techniques used in hydrology and applying them to glaciology, something which can greatly benefit the field.

However, I don't find the way in which the paper is currently written to be suitable for the target audience of a journal such as The Cryosphere. Many of the terms and methods which may be more familiar to trained hydrologists will not be familiar to many glaciologists. I would very much like to see this paper published in The Cryosphere as it would be a good step towards better collaboration between these fields, but I would recommend a detailed re-write to make it more accessible to those without a background in hydrology.

*We thank the reviewer for this perspective, and indeed our collaboration of a hydrologist (lead author) and Greenland specialists (most other authors) resulted in many rounds of internal editing and terminology clarification. We targeted TC for this submission because we felt it was most novel and interesting to the glaciology community, and we are glad the reviewer agrees with that decision.*

*We have made many changes to the text to better define jargon and standard practices from hydrology for the glaciology community, and the reviewer's detailed comments where helpful in guiding where these passages might be needed. When a full and satisfying explanation might have taken too much text In making these changes, we have referred readers to classic citations (while still including hopefully enough information here so the reader doesn't have to look it up). Our changes are highlighted as responses to the specific concerns below.*

My detailed comments are below:
Line 34: and lakes?
*We sought to define and distinguish the emerging rivers and streams research against the more established study of lakes here, although we agree that the study of supraglacial hydrology as a whole is an emerging discipline. The paragraph beginning on line 61 covers relevant lake research, including some new papers suggested by both reviewers (see below).*

Line 40: Also Leeson et al. 2012 DOI: 10.5194/tc-6-1077-201
*Cited as suggested by both reviewers. Thanks for noting this omission.*

Line 82: What about subglacial channels/ transport through firn?
*We have included the parenthetical "(which include firn atop the GrIS)" in this line to establish that 'hillslope' soil processes are akin to firn/bare ice process in Greenland.*

Line 97: Is this the only data available for the whole of Greenland? It seems a big
weakness of this paper is the lack of validation data available. If other datasets were
not used it should be explained why (but if this is really all there is then fair enough!).

*This is, to our knowledge, the only validation available in the whole of Greenland. These data are very
difficult to obtain in the field, and while there are a few other observations of river discharge atop the
GrIS (discussed in lines 122-125), the Smith et al data are the only available source sufficient for
calibration of a hydraulic model as performed here. We agree that this is a weakness of this paper, but
hope that this paper spurs other authors (and funders) to collect more in situ flow records to push this
research forward.*

Line 122: Related to the previous comment, is there data for other times available?
Can you explain why it is so important only the peak of the melt-season can be used?
*We see that our writing was confusing! It is not necessary that peak melt season data are used, and the
methods should work at any time. Per previous work, we expect peak melt season to correspond to the
maximum drainage network extent and therefore the most efficient drainage of the ice sheet (and thus
most likely to agree with SMB modelling). We have eliminated this phrase to avoid similar confusion.*

Line 129-133: Were other images available? What if other streams are present and
just didn't have water flowing on this particular day?
*The stream network from Smith et al (2017) was investigated by Yang et al (2018) as cited, and that
work showed that the peak melt season image we re-use here produced the most extensive stream
network and that channels were visible in the image. Since we couple this image to the DEM, and the
DEM is ultimately the product that determines how many streams are present, we are not reliant on the
representativeness of this single image as the DEM should generate channels not captured in the image.
That said, and toward a question the reviewer asks later, the narrowest streams here are ~0.2m, and
thus this 0.5m imagery is not missing any features that would have a major effect on flow routing.*

*This passage has been edited as follows: "Smith et al.'s stream network product was combined with a
seasonally simultaneous portion of the 2 m resolution ArcticDEM DEM (Digital Elevation Model) obtained
from Polar Geospatial Centre (Porter et al., 2018) to produce two distinct supraglacial stream networks
as described in Section 3.2."*

Line 134: Please define DEM
*Defined as requested*

Line 142: How do you determine all of the water in each of these grid cells will remain
in this catchment, could any of it be being transported elsewhere?
*This is an excellent question. Terrestrial hydrology theory holds that all water at the surface or in the
near surface (in the GrIS- in a river, lake, or the firn/crust) is topographically controlled. That is, the
watershed as defined by elevation by definition contains all transport as water flows 'down-gradient'.
This assumption could be incorrect in several cases that are not present in terrestrial hydrology: 1) if
water penetrates the firn into an englacial system driven by pressure heads and not topography, 2) if the
channel network in reality thermally erodes some topography to the point where it 'breaches' the
watershed divide, or 3) if there are large scale subsurface fluxes that transport subsurface water into or
out of the system. In terrestrial hydrology, these subsurface fluxes do exist, leading to discussions on the
'myth of the water balance' in the literature. However, for scales such as the catchment here these fluxes
are extremely uncommon. In the absence of literature describing englacial fluxes at medium to GrIS*

scales (which in fairness would be extraordinarily difficult to measure), we are confident in this classical assumption and believe the 3$^{rd}$ exception is not valid. Further, the remotely sensed image of streams shows the stream network as it actually exists (as opposed to the traditional hydrologic view driven solely by a DEM that would assume topography), and therefore we can eliminate the 2$^{nd}$ exception. We are left with the 1$^{st}$ exception as a plausible means to violate our assumption of topographically controlled transport.

To state this assumption formally, we add the following to this passage: "which we assume is topographically constrained and transported exclusively via surface/near-surface transport" . We also state (per other suggestions by both reviewers) the following: "Further, the use of a single R$_{coef}$ allows us to accurately model discharge without allowing attribution of errors in runoff production: these could stem from SMB errors, unaccounted for refreezing, storage, or lake filling, surface transport that violates topographic constraints, englacial draining, or ADCP measurement error. Our framework is unable to apportion any gaps in runoff production and routed discharge to any of these sources, and thus our treatment of runoff as a bulk reduction/augmentation is faithful to our experiment design and manuscript goals."

Line 167: Please define Froude number.
*The Froude number is a classic hydraulic ratio of velocity and celerity in an open channel. The Froude number determines the flow regime (sub critical, critical, or supercritical), and these regimes are used in canal and river restoration design. Froude numbers essentially indicate whether surface waves propagate upstream or downstream given the balance of gravity driven flow and energy driven flow from upstream.*

*Per the reviewer's suggestions to improve the use of hydrologic jargon, we have added "(a classic index of flow velocity in open channel hydraulics)" after the introduction of the term.*

Line 163 onwards: Why was none of this data suitable to use to validate the model?
*Great question. In order to uses these data to validate the model, we would have to set it up in a controlled experiment: we would put the correct runoff forcing upstream of only these measurements and then route just the appropriate subbasin to produce the model estimates of these hydraulics at the appropriate location. We could have used these measurements if they were made at the same location at the ADCP. The internal model calculations that produce the hydraulics must by definition match these measurements at a point given conservation of mass if the input is correct, so this poses a problem given the lack of process modelling and mass gaps we discuss extensively later.*

*In further service of jargon translation, we now close this paragraph with "Note that we cannot use these observations to validate our routing model, and instead use them to inform it. These point measurements could in theory be reproduced by our hydraulic model, but to do so would require measurements of channel properties and runoff upstream of each point for several hours/days before each hydraulic measurement was taken, and such data do not exist."*

Line 176: I know what you're getting at here but I'm not sure 'factorial' is the correct description?
*Since this has sown confusion, we have dropped this and instead state the simpler ("four runs per model") in this passage.*

Line 185: Is it possible that the SMB values are correct and water just isn't making it to

the channels where the ADCP measurements are made? E.g. What about refreezing in firn?

*This is possible, but we cannot say for sure, and it would seem a large amount of mass to refreeze. It could also be that a slowly draining englacial system is in play, or refreezing in the bottom of a lake, but we have seen no evidence of these factors in this watershed. It is true that we observe less outflow than SMB runoff would predict. Since we do not know why this might be the case, the use of $R_{coef}$ allows us to account for this difference without attribution. We do not want to make any conclusions we cannot support, and our modelling setup does not let us state anything beyond what we observed- there is a mismatch between how much mass is produced and how much is conserved to pass the outlet with correct timing. Reviewer 2 had similar concerns (although their focus on was lake filling and englacial drainage), so we have substantially rewritten the discussion and methods. We copy the new text below*

*We now state in 3.1: "Further, the use of a single $R_{coef}$ allows us to accurately model discharge without allowing attribution of errors in runoff production: these could stem from SMB errors, unaccounted for refreezing, storage, or lake filling, surface transport that violates topographic constraints, englacial draining, or ADCP measurement error. Our framework is unable to apportion any gaps in runoff production and routed discharge to any of these sources, and thus our treatment of runoff as a bulk reduction/augmentation is faithful to our experiment design and manuscript goals."*

*And in the discussion [entire paragraph reprinted]:*
*        "Our results also support earlier assertions of mismatched timing and magnitude of SMB runoff and observed discharges entering the Rio Behar terminal moulin (Smith et al., 2017).  The routing model is unable to assign glaciologic process to mass gaps, so we can only suggest plausible mechanisms for closing that mass balance gap. Mass gaps could perhaps result from to subsurface retention and/or refreezing in bare-ice weathering crust (Cooper et al., 2018), a process not currently well-represented in SMB models. Or, the mass imbalance could come from transport processes: filling lakes, drainage through fractures (there are no crevasses in the study area), or the breach of topographic divides are all plausible transport process gaps.  Topographic breach is unlikely given that we use an observed (via image) channel network, and thus if breaches did occur they are accounted for. Further, total depression storage (including true lakes and DEM artifacts) was $6.92e^6$ $m^3$, which is two orders of magnitude less than the observed ADCP flux during this time (integrated into a bulk volume, $241e^6$ $m^3$) and one order of magnitude less than the maximum runoff deficit (obtained by subtracting the ADCP from the largest SMB input, $17.5$ $e^6$ $m^3$).  Therefore, if all depressions were dry at the start of routing and were completely filled by runoff before beginning to flow in the channel network, this would still only account for roughly one-third of extra runoff production mass. Given that we know lakes are full during this time period, we assert that this lake filling effect is not the cause of mass imbalance. Further, errors in our outlet hydrographs are dominated by underestimation of night-time low flow periods, as peak flows are modelled well across nearly all 16 trials.  These night-time low flows are particularly important for mass balance in the Rio Behar watershed, as a large driver of mismatches in total mass balance (Figure 4) comes from these low flow periods. Error could come from the ADCP itself, and this instrument is generally less certain at lower flows. However, the ADCP record here is taken from Smith et al. (2017) and represents a well documented procedure carried out by expert field personnel, and thus we are confident that ADCP errors are too small to explain $R_{coef}$. We affirm that all SMB models examined here produce too much excess water relative to ADCP observations (at least at peak times, Figure 4 shows MERRA2 total runoff is less than the ADCP total discharge, but still requires $R_{coef} < 1$ to reduce the peak daytime volume of water), and do not model night-time flows without routing, consistent with Smith et al. (2017). Our results suggest that hydrologic process modelling (i.e., routing) can correctly reproduce these night-time low flows."*

Figure 1: The middle 4 squares could be laid out in a clearer way. Why does including or excluding hillslope flow have a 'yes' or 'no' but there is no differentiation between coarse and fine network?
*Good question. We have streamlined the figure as follows, which hopefully communicates more clearly.*

[Figure]

Line 195-6: 'generalizing the process of water routing from satellite image collection to water routing' doesn't make sense to me.
*Excellent point, as this was written in a way that doesn't make much sense upon rereading, and Reviewer 2 was also confused. We meant that we wanted to build a general and flexible process that can start with a remotely sensed channel map and take that map all the way to hourly routing (given calibration data). This passage is now stated "Although Smith et al. (2017) provide a topologically connected channel network for our study area (i.e., they explicitly defined how every channel is connected to every other channel throughout the entire network to allow water to flow from the headwaters to the outlet to obey observed channel connections) , we are interested in generalizing the process of water routing from satellite image collection to water routing in cases where pre-existing channel network maps do not exist., Further, we must generate different river networks to which is also necessary to test the effects of network density on the routing model."*

Line 195 onwards: This paragraph was one I especially felt could be better explained for the non-hydrologist. It would also be good to see justification for 'standard hydrologic' practices being applied to icy surfaces.
*Understood! We have added hopefully clarifying passages that have made this paragraph quite a bit longer (we split it into multiple paragraphs), but hopefully more legible. Since this was a major point of confusion, we have drafted revised text and reprint it here:*

[revised manuscript text omitted]

Line 210-11: 0.5 is more than twice 0.2 so I wouldn't say they are matching, does this difference matter?
*Thanks for pointing it out- what is important is the order of magnitude matching, as the differences between a stream of 0.2m and 0.5m are negligible for routing. Also, since remotely sensed stream widths are directly appended to the DEM generated channels (see updated text quoted above), this means simply that the process is capturing streams down to the limit of the imagery, which is a positive result.*

Line 217 onwards: Again this paragraph could be clearer. The authors discuss stream order again in the results so an introduction to what this means and why this matters would be helpful.
*Great suggestion. We have now included citation to a classic reference with explanation : "The coarse network has six stream orders (e.g. the smallest streams on the landscape are defined as order 1, and every junction of stream produces a new stream of higher order), and the fine network seven orders. Stream orders are a shorthand for hydraulic complexity of a network as the number and length of streams in a given order both increase geometrically (Horton, 1945). Therefore, our finding of almost an order of magnitude more channels in the fine 7-order network than the course 6-order network matches theory"*

Line 234: Do any other models have this rigorous representation? If they are written in other, more accessible, languages it may still be useful to briefly mention them for those who may want to build on this but not use FORTRAN.
*There are essentially only two options for efficient flow routing 'off the shelf' at large scales like these in current terrestrial hydrology: HRR, and RAPIDE (David et al., 2011). RAPIDE uses Muskingum routing instead of Muskingum-Cunge routing and does not contain a hillslope model, which makes it inferior for this application as we can't simulate hillslope (e.g. firn and bare ice) flow separately per subcatchment. In our view, the use of FORTRAN is a strength- it is an open source language that computes very quickly. If we were to recode HRR in Python, R, or Julia (considering other common open source languages), it would run orders of magnitude more slowly. We wrap HRR in an R framework- we invoke its FORTRAN core from RStudio, which allows for an easily shared workflow.*

Line 245: Please give a little more detail about how EAs work.
*We want include the intent of the reviewer without diving too deeply into the very large literature on hydrologic model calibration, so we have included the following passage "A very large literature on hydrologic model optimization and calibration exists, and interested readers are referred to Kirshner (2006) and Gupta et al (1998) for broad overviews of the subject. We do perform calibration this using an established evolutionary algorithm (EA; NSGA II, Deb et al., 2002) as EAs are efficient estimators in large parameter spaces that can achieve near-optimal results (Gleason and Smith, 2014)." The citations we give here are classic point and counterpoint papers that approach the subject of model calibration in a high level discourse. The Gupta school speaks to the art of calibration and the ability to improve*

*outcomes, while the Kirshner school bemoans that model calibration has become an art unto itself that obscures true understanding.*

Line 255: Please define Manning's n.
We have added the following text in this passage:
" *Channel friction is represented by Manning's* n *and the EA solves for a single n per bin and assigns that n to all streams falling within that drainage area threshold. Manning (1891) generalized open channel flow into a simple equation where all flow resistances are lumped into a single empirical parameter n, and over a century of subsequent research has related n to landscape variables, channel form, and other geomorphic controls.   Our binning of Manning's* n *follows general hydraulic correlations between channel size, slope, total discharge, and* n *(Brinkerhoff et al., 2019)."*

Line 296-7: Please define acronyms.
*The hydrological acronym has been previously defined (NSE in line 275).*

Line 311 and figure 4: Please comment more on why we are assuming here that
MERRA2 is incorrect. Could it not be that all the other inputs are less accurate and
MERRA2 is actually getting it right?
*We didn't assume any models were correct/incorrect in this study in the beginning, but previous literature suggested we might need to adjust runoff. We took the published SMB output and compared it to field measurements before and after routing, and used the parameters we solved for (especially $R_{coef}$) as a guide for how much correction each model requires to match observations. We do assume that our ADCP is correct, but this doesn't mean that the SMB model is incorrect: there is a process gap between what the models produce and what we observe, and this gap cannot be closed by flow routing. That gap is highlighted for the first time in this manuscript.  MERRA2 has the same timing errors as all the other models, although it has a much better order of magnitude of flow. However, while MERRA2 is the closest 'out of the box' it is not the most accurate model after routing, and ultimately none of the models are correct in matching the ADCP without routing.*

Line 402 onwards: How is the slow lateral transport accounted for in the model?
*Lateral transport is accounted for as hillslope flow. A true physical model would attempt to account for all of the processes we know occur in firn/crust transport to channels, but since that knowledge (and observations) is not sufficient to parameterize such a model, we use the hillslope friction coefficients as a bulk control on the speed of water as it moves over/through the ice/crust/firn and into channels. We now state the following to avoid confusion "That is, since we lump all flow over/through the ice/firn/snow/crust before it reaches channels into a single 'hillslope' flow with a single friction, we can be confident in the speed of this transport but not its flowpaths or mechanism."*

Line 420: Can you give an example(s) of the physical processes that may be leading
to this to support this conclusion?
*We see how this was confusing, as the use of positive/negative sentence construction was poor. We now state more plainly "However, we believe that model calibration statistics at the outlet indicate the physical realism of the process we're attempting to model: since we modelled an accurate outlet hydrograph, the fully mass and momentum conserved physics of HRR mean that upstream flows must be realistically represented or we could not have produced a quality outlet hydrograph."*

Line 443: Anywhere or just in the bare-ice ablation zone?

*We believe our study can be repeated anywhere there is a stream network, regardless of the glaciologic regime. In practice, stream networks form most often in this zone, but the physics of open channel flow will be the same in all networks across all settings. The parameters will change to yield different flow velocities through hillslopes and channels, but the basic premise will hold.*

---

## Author Comment (AC2) · 10 Feb 2021

*Below, find the original reviewer concern/question in regular typeface followed by our response in italic typeface. This response is written in anticipation of making all corrections to a manuscript as stated. Therefore, we make statements such as 'we changed' or 'we corrected' or 'we modified' in past-tense as if this document were being submitted together with a revised manuscript. It is our intent that the revised manuscript will read exactly as suggested here. We have also included references to the other review.*

This is a novel piece of research addressing an important area of glaciology. It uses a well-established 'off the shelf' hillslope-channel hydrological model used previously in terrestrial settings, to route meltwater across a supraglacial catchment on the Greenland Ice Sheet. It provides a useful steppingstone, therefore, to one day developing fully coupled surface mass balance – surface water routing – subglacial water routing – glacier dynamics models. The paper adds to valuable previous work produced by this group. A series of experiments are created using four different runoff series from widely cited surface mass balance models (HIRHAM5, MAR3.6, RACMO2.3, and MERRA-2) and for routing model sets ups that consider a course and a fine density channel network, and which either consider or do not consider hillslope routing processes. So 16 separate experiments are performed in total. Model parameters are calibrated by comparing outputs with measured discharges in a surface stream at the catchment outlet over a few days in July, which have been published elsewhere. Parameter space is explored, and the patterns of parameter values are used to infer the importance of catchment processes, for example the role of hillslope friction, which is high compared to terrestrial settings and which, it is argued, may represent shallow subsurface routing through a weathering crust.

The paper is nicely structured and generally well written, and the Figures and Tables are clear and useful. The work is thorough, generally acknowledges previous work (with a few exceptions that could be added at the authors' discretion – see below) and provides a valuable contribution to the literature.

*Thank you for this assessment! We appreciate this summary and have agreed to functionally all of your suggestions below. Thanks for taking the time to write all these down and read the paper in detail, as it is much improved because of both reviewer's efforts.*

There are five places where I think things need to be explained in more detail or where the results could be discussed further. These are:
1. On line 210/11 you tell us the channel widths that are produced for the fine network. But how are channel widths determined? Was this explained? I'd assumed they'd be dictated by the DEM grid size? Why don't you also tell us the widths of the channels produced for the coarse network here? [*note the reviewer bumped this comment from detailed to broad, so we include some details that respond to later requests here*].

*Thanks for pointing this out, as we did neglect to mention this important methodological procedure. Channel widths were set by the remotely sensed image, and therefore they are limited to the resolution of that imagery. This is because we 'burned in' the network, that is, we took Smith et al's stream map and lowered it into the ice to help the DEM-based network generation. Therefore, once the DEM (with burned in stream map to guide it) created either fine or coarse networks, we assigned the width of the nearest remotely sensed channel from the Smith map to the DEM-generated channel. In practice, this*

*means that channel widths are identical across networks for higher order (larger) channels as these appear in both networks. The only real difference between the networks is at lower orders, and in this case we are confident in the widths as the DEM based method will always co-locate a topographic channel with the RS channel (due to its artificially lowered elevation from burning in), and thus they are accurate to the resolution of the network. Therefore, the smallest streams may be overestimated in width, but our field observations in previous papers indicate that the very smallest true open channels are on the order 0.2m width, and the imagery produces streams to 0.5m (as we note in the paper).*

*Thus, to your point, the streams are limited by image resolution, not DEM resolution.*

*Reviewer 1 also had questions here, so we substantially re wrote this passage. We reprint it in full below in response to another of your questions, but the germane text to this specific query now reads as follows: "To estimate the impact of drainage pattern on meltwater routing, we tested both a large ($10^4$ $m^2$) and a small ($10^3$ $m^2$) channel initiation threshold to create a 'coarse' and a 'fine' supraglacial drainage network, respectively from the DEM (Figure 2). These two modelled stream networks both follow the channel map from Smith et al (2017), with the key difference that the coarse network does not produce the narrowest streams we know to exist. This enabled us to test the effects of including or excluding very small tributary streams on surface water routing. We assign channel widths to each DEM-derived channel from the Smith et al channel map, and since the DEM process begins with burning in these streams, there is always a 1:1 assignment of channel width from imagery to network model. Our 'fine' channel network produces streams with a minimum width of 0.5m, matching to the correct order magnitude Gleason et al.'s (2016) reporting of channels as narrow as 0.2m. The coarse network produced streams with a minimum width of 0.7m, suggesting it is excluding the smallest streams in the remotely sensed map."*

2. On line 222. You say "and lakes are represented by wide, shallow 'throughflow' river segments: : :" Is this a major limitation of this work? How do lakes fill and drain? Surely, the filling and draining of lakes will have a major impact on the relation between distributed runoff and the hydrographs at the catchment outflow and yet this important process is not incorporated. I'd like to see more discussion of this. How many sinks are there in the catchment that need to be filled? Where are they? What are their volumes? What are the implications for water routing? [*note this comment was bumped from detailed to broad, so we include some details from later detailed requests*].

*These are excellent questions, and answering them improves the manuscript. Terrestrial hydraulic routing as a whole typically incorporates lakes in the manner we have treated them here. For a free-flowing natural lake connected to the network, treatment of these lakes as a wide reach with low slope is sufficient to accurately reproduce the behavior of these fluvial lakes. At-scale terrestrial hydrology struggles much more with reservoirs: places where the mass balance of a lake is affected by human decisions that are very difficult to model.*

*We conceive of connected lakes on the GrIS as closer to fluvial throughflow lakes than reservoirs: they fill during the early season until an outlet sill elevation is reached, and then these lakes spill into the downstream channel, furthering the network connection and extent. Thus, during peak flow and maximum network extent (as for the data in this paper) we are relatively confident that our treatment of connected lakes is properly conserving mass into and out of these systems. Further, although our methods cannot explicitly fill a lake with full consideration of bathymetric geometry, we do see that at times during the routing the outflow from a lake segment is effectively zero (less than 0.0001 $m^3$/s) due*

to the small amount of inflow and hillslope input to the lake. The lake segment does not dewater, but rather stores water with this effectively zero output. Thus, the model is able to show there are times that the lake is connected to the network but not actively spilling until flow returns at a higher level the next day, effectively segmenting the network. This effect is also a function of lake order- if a lake is on the first order (headwater) this spill/no spill condition is much more likely to occur than in a 5th or 6th order lake that collects water from the entire network.

Later in this review, you note that some of our slow transport over hillslopes might in fact be due to unaccounted for lake filling. This is plausible, but unlikely for this peak-melt season experiment as all lakes are full here, and if they do stop spilling the model indicates this only happens overnight. Repeating the experiment with a less developed river network built from an early-season image and coupled with early-season SMB output and some observations of lake filling would allow us to assess this explicitly. Or, we could use a validated lake filling model from the literature and try to harmonize the forcing to see if our lakes spill at the same time as that model.

Finally, per another request made below, we have calculated the total depression storage in the basin to assess the potential magnitude of their effect on mass imbalance. We see in the following table that the total depression storage (including DEM artifacts/errors in addition to lakes) is $6.92e^6$ m$^3$, which is two orders of magnitude less than the observed ADCP flux during this time (integrated into a bulk volume, $241 e^6$ m$^3$ ) and one order of magnitude less than the maximum runoff deficient (obtained by subtracting the ADCP form the largest SMB input, $17.5 e^6$ m$^3$).  Therefore, if all depressions were dry when we started and all the extra runoff filled them, this would still only account for roughly one-third of the missing mass. Given that we know lakes are full and therefore cannot fill from empty, we assert that this lake filling effect is not the cause of mass imbalance.

| | Total depression storage | ADCP total | Minimum SMB | Maximum SMB | Maximum runoff deficit |
|---|---|---|---|---|---|
| Volume (m3) | $6.92 e^6$ | $241 e^6$ | $219 e^6$ | $417 e^6$ | $17.5e^6$ |

To capture these ideas and present them to readers, we have added the following text here: "The main trunk streams only are visible in the coarse network, and lakes connected to the channel network (i.e., have an inflow and outflow) are represented by wide, shallow 'throughflow' river segments as all are non-terminal with outflow channels. Lakes on the GrIS evolve seasonally- they begin pooling water in the early melt season until an outlet elevation is reached, and then they begin to spill downstream. Our data come from peak melt season when lakes are full, and thus any lake connected to the network will behave fluvially, that is, it will spill according to its slope, volume, and lateral input via the conservation of mass and momentum. Further, Figure 2 indicates that there are likely no lakes in the watershed that are disconnected to the channel network- our drainage density is sufficient to ensure that lakes of any appreciable size would be captured as a throughflow segment."

Further, we added the following in the discussion  "Or, the mass imbalance could come from transport processes: filling lakes, drainage through fractures (there are no crevasses in the study area), or the breach of topographic divides are all plausible transport process gaps.  Topographic breach is unlikely given that we use an observed (via image) channel network, and thus if breaches did occur they are accounted for. Further, total depression storage (including true lakes and DEM artifacts) was $6.92e^6$ m$^3$, which is two orders of magnitude less than the observed ADCP flux during this time (integrated into a

*bulk volume, 241e$^6$ m$^3$) and one order of magnitude less than the maximum runoff deficit (obtained by subtracting the ADCP from the largest SMB input, 17.5 e$^6$ m$^3$). Therefore, if all depressions were dry at the start of routing and were completely filled by runoff before beginning to flow in the channel network, this would still only account for roughly one-third of extra runoff production mass. Given that we know lakes are full during this time period, we assert that this lake filling effect is not the cause of mass imbalance."*

3. On line 397 you show that after calibration, hillslope friction values were on occasion very high – up to the max. threshold allowed of 25. These do seem very high. To what extent are these high because you didn't allow channel frictions to rise very high (Table 2)? Could you explain a bit more about the range of parameter values considered in Table 2 and the implications of this?
*This is another good observation. You are spot-on that an unconstrained model can effectively 'trade off' channel friction and hillslope friction to produce a single effective friction for the stream unit. We therefore need to constrains these parameters to realism. Manning's n has a history of study in ice channels and we have a good sense from previous literature about what range of values are plausible for this parameter. We have almost no information on effective friction for ice/firn hillslopes that funnel this water to channels. Therefore, we limited channel friction to what the literature indicates is physically possible and let the model choose hillslope friction from a wide range of values: We are comfortable limiting Manning's n to achieve this given its over 100 years of study. Our results indicate that a considerable amount of total friction is needed: Figure 7 shows that non-hillslope models need extreme channel friction and do not model outflows as accurately as models with hillslopes to add friction. Thus, the mass and momentum conservation inherent to our setup indicates that water needs be slowed, and the most physically realistic way to slow it is via a fine network with hillslopes included, which matches theory and observation.*

*We have added the following text in the following paragraph to help tease this point out:*
*"HRR is not a glaciological model, and therefore it is agnostic about sources of friction and can trade off channel and hillslope friction to produce correct outflows if unconstrained. We have constrained the channel friction to match literature field observations closely and allowed hillslope frictions to vary over a much wider range of values given the longer history of study and larger databases of Manning's n values for ice channels relative to transport through the crust/bare ice. Therefore, non-hillslope models would likely improve only by including physically unrealistic channel frictional values given results in Figure 7."*

4. Related to point 2 above. On lines 398-405, you find very slow water transport rates on hillslopes here and suggest this may be due to slow transport through an ice crust. But what role does ignoring travel through lakes as they fill up and overtop their outlet channels play on your results?
*Per the above, this concern arises from unclear writing on our part. We are confident that lakes are handled with realistic 'fill and spill' behavior, particularly for this fully developed network from July.*

5. You need to introduce a runoff correction coefficient (Rcoef) to match modelled with measured hydrographs and this turns out in the calibration to always be less than one which means runoff from all the SMB models is over predicted or measured discharge at the catchment outlet is under predicted. This is discussed to some extent in the Discussion but I think more thought could be given to this. You suggest water may be stored (and possibly freeze) in a weathering crust but is this feasible? What volumes are we talking about and could storage in a weathering crust really explain it? Related

to above, is it possible lakes may be filling over this period which would explain the discrepancy. Could it also be due to leakage into the ice sheet via crevasses?

Other than these five points, I have just a series of questions / suggestions regarding improving clarity as follows.

*Reviewer 1 is also interested in this mismatch. In writing about our results, we want to be careful about only stating supported conclusions and stating plausible hypotheses specifically as plausible but not supported conclusions. Our results show (as you point out) that $R_{coef}$ is always less than 1, and therefore we have to both slow down water (channel and hillslope friction) AND hold 40-60% of it within the watershed in order to match flows. There are several possibilities for this underestimation: [note this text is identical to our response to reviewer 1 in quotes] "This assumption could be incorrect in several cases that are not present in terrestrial hydrology: 1) if water penetrates the firn into an englacial system driven by pressure heads and not topography, 2) if the channel network in reality thermally erodes some topography to the point where it 'breaches' the watershed divide, or 3) if there are large scale subsurface fluxes that transport subsurface water into or out of the system. In terrestrial hydrology, these subsurface fluxes do exist, leading to discussions on the 'myth of the water balance' in the literature. However, for scales such as the catchment here these fluxes are extremely uncommon. In the absence of literature describing englacial fluxes at medium to GrIS scales (which in fairness would be extraordinarily difficult to measure), we are confident in this classical assumption and believe the 3$^{rd}$ exception is not value. Further, the remotely sensed image of streams shows the stream network as it actually exists (as opposed to the traditional hydrologic view driven solely by a DEM that would assume topography), and therefore we can eliminate the 2$^{nd}$ exception. We are left with the 1$^{st}$ exception as a plausible means to violate our assumption of topographically controlled transport." Our ADCP measurements could also be error, but ADCP error seems the least likely as a mature technology applied by an expert team with thorough control (although we show our bias as field hydrologists with that statement!). In response to this and Reviewer 1's comments, we have added the following to the paper in section 3.1*

*"Further, the use of a single $R_{coef}$ allows us to accurately model discharge without allowing attribution of errors in runoff production: these could stem from SMB errors, unaccounted for refreezing, storage, or lake filling, surface transport that violates topographic constraints, englacial draining, or ADCP measurement error. Our framework is unable to apportion any gaps in runoff production and routed discharge to any of these sources, and thus our treatment of runoff as a bulk reduction/augmentation is faithful to our experiment design and manuscript goals."*

*And in the discussion [entire paragraph reprinted, which includes previously quoted passages]:*

*"Our results also support earlier assertions of mismatched timing and magnitude of SMB runoff and observed discharges entering the Rio Behar terminal moulin (Smith et al., 2017). The routing model is unable to assign glaciologic process to mass gaps, so we can only suggest plausible mechanisms for closing that mass balance gap. Mass gaps could perhaps result from to subsurface retention and/or refreezing in bare-ice weathering crust (Cooper et al., 2018), a process not currently well-represented in SMB models. Or, the mass imbalance could come from transport processes: filling lakes, drainage through fractures (there are no crevasses in the study area), or the breach of topographic divides are all plausible transport process gaps. Topographic breach is unlikely given that we use an observed (via image) channel network, and thus if breaches did occur they are accounted for. Further, total depression storage (including true lakes and DEM artifacts) was 6.92e$^6$ m$^3$, which is two orders of magnitude less than the observed ADCP flux during this time (integrated into a bulk volume, 241e$^6$ m$^3$) and one order of magnitude less than the maximum runoff deficit (obtained by subtracting the ADCP from the largest SMB input, 17.5 e$^6$ m$^3$). Therefore, if all depressions were dry at the start of routing and were completely filled*

*by runoff before beginning to flow in the channel network, this would still only account for roughly one-third of extra runoff production mass. Given that we know lakes are full during this time period, we assert that this lake filling effect is not the cause of mass imbalance. Further, errors in our outlet hydrographs are dominated by underestimation of night-time low flow periods, as peak flows are modelled well across nearly all 16 trials. These night-time low flows are particularly important for mass balance in the Rio Behar watershed, as a large driver of mismatches in total mass balance (Figure 4) comes from these low flow periods. Error could come from the ADCP itself, and this instrument is generally less certain at lower flows. However, the ADCP record here is taken from Smith et al. (2017) and represents a well documented procedure carried out by expert field personnel, and thus we are confident that ADCP errors are too small to explain $R_{coef}$. We affirm that all SMB models examined here produce too much excess water relative to ADCP observations (at least at peak times, Figure 4 shows MERRA2 total runoff is less than the ADCP total discharge, but still requires $R_{coef} < 1$ to reduce the peak daytime volume of water),and do not model night-time flows without routing, consistent with Smith et al. (2017). Our results suggest that hydrologic process modelling (i.e., routing) can correctly reproduce these night-time low flows."*

L54. After 'lake impoundment' you could consider referring to: Arnold, N.S., Banwell, A.F. and Willis, I.C., 2014. High-resolution modelling of the seasonal evolution of surface water storage on the Greenland Ice Sheet. The Cryosphere.
L63/4. As well as Banwell et al 2013, could also refer here to Banwell, A., Hewitt, I., Willis, I. and Arnold, N., 2016. Moulin density controls drainage development beneath the Greenland ice sheet. Journal of Geophysical Research Earth Surface, v. 121, p.2248-2269.
*Both papers cited as suggested.*

L64. Immediately before the sentence beginning 'Liston and Mernild (2012)' you could add another sentence summarising and referring to the work of Leeson et al 2012: Leeson, AA, Shepherd, A, Palmer, S, Sundal, A & Fettweis, X 2012, 'Simulating the growth of supraglacial lakes at the western margin of the Greenland ice sheet', Cryosphere, vol. 6, no. 5, pp. 1077-1086.
*As requested, we now state the following here "Leeson et al (2012) similarly used Manning's equation to transport water in a 2D grid based routing scheme, assigning all grids a uniform Manning's n while not explicitly defining flow differences between flow in channels or flow over bare ice."*

L75-77. Would it be better to say: "These previous efforts successfully modelled meltwater transport on the GrIS ablation zone, but their relative simplicity allows space for more sophisticated routing models from terrestrial hydrology to be applied to ice sheet surfaces"? Or something like this.
*It would be better to say that! Thanks for the suggestion.*

L85. What is meant by "explicit routing"? As opposed to implicit routing? What about saying "...accurately routing water at the catchment scale across the GrIS"?
*This now reads "barriers to applying such routing"*

L86. What is meant by "explicit" here?
*Explicit here means that each member of the network has all of its attributes defined.*

L87/88. ": : :but to our knowledge no automated, large scale network extraction and

topological connection work exists for the GrIS". Do you mean for the whole of the GrIS here ? There are examples of this at smaller scale surely.
*We meant that we are unaware of any off the shelf processes that both extract a network and define its topology at scale (e.g. for channel networks of thousands of members or more). We have added the word 'coupled' to this passage to indicate this.*

L89. "these issues" What issues? You've not explicitly referred to 'issues' above.
*Changed to "yet these cannot be applied until a generalizable automated extraction/topological connection process is available" to clarify.*

L92. What are "network scales"? You've referred to catchments and (I think) the entire ice sheet previously which I understand but what are these?
*'Network scales' channel networks of thousands of members or more. We realize now we've had that definition in our head and never written it in this manuscript. So, we have added the epithet (i.e. catchments with thousands of channels or more) earlier in this paragraph.*

L93. Delete the word "these" as you've not referred to processes previously.
*We are here referring to hillslope friction, channel friction, channel density, runoff transport, and runoff reduction/augmentation as introduced in this sentence.*

L96. What is meant by "spatially explicit"?
*In terrestrial hydrology, routing networks can often be spatially implicit, and a system represented as a series of links and nodes that do not necessarily map precisely back to reality. In our case, each of our channels can be mapped to its physical location on the ice sheet.*

L104. Would it be clearer to say ": : :test how the representation of hillslope processes and network density (as derived by our automated network generation process): : :" or something like that?
*Great suggestion. Accepted.*

L116-119. Would it be clearer to say: "Previous work in the basin includes: i) a comparison of SMB runoff and field measured discharge using a simple routing method (Smith et al., 2017); ii) a study of subsurface water storage in bare-ice weathering crust (Cooper et al., 2018); iii) albedo mapping (Ryan et al., 2017); and iv) mapping the catchment's supraglacial channel network via satellite and un-crewed aerial vehicle (UAV) remote sensing (Ryan et al., 2017; Yang et al., 2018)"?
*Another good idea. Accepted.*

L122. 'of a summer'
*No longer relevant due to a change requested by Reviewer 1*

L128/9. Perhaps add Banwell et al 2016 to this list?
*Cited as suggested*

L135. Could reference Pope et al, and Moussavi et al somewhere here as they use the ArcticDEM in the context of representing GrIS surface lake bathymetries
Pope, A., Scambos, T. A., Moussavi, M., Tedesco, M., Willis, M., Shean, D., and Grigsby, S.: Estimating supraglacial lake depth in West Greenland using Landsat

8 and comparison with other multispectral methods, The Cryosphere, 10, 15–27, https://doi.org/10.5194/tc-10-15-2016, 2016.
Moussavi, M. S., Abdalati, W., Pope, A., Scambos, T., Tedesco, M., MacFerrin, M., and Grigsby, S.: Derivation and validation of supraglacial lake volumes on the Greenland Ice Sheet from high-resolution satellite imagery, Remote Sens. Environ., 183, 294–303, https://doi.org/10.1016/j.rse.2016.05.024, 2016.
*Cited as suggested*

L138-143. Can you better explain here how you deal with the very different grid sizes between the 2m DEM and the various (I assume) grid sizes of the 4 models? How do you interpolate across the finer scale grid? Is the Rio Behar catchment entirely within a SMB model grid?
*At 63km$^2$, Rio Behar covers a small amount of ground area relative to SMB cells. HIRHAM5 has 5.5km cells (30.25km$^2$, 8 cells intersecting), MAR 3.6 has 20km cells (400km$^2$, 2 cells intersecting), RACMO2.3 has 11km cells (121km$^2$, 3 cells intersecting), and MERRA2 0.5degree cells (~58 x 30km here, 1740km$^2$, entirely contained within 1 cell). Thus, our domain takes 1-8 grid cells of the original models, and straightforward averaging is effective, per Smith et al., 2017. We have added the following in this passage:*

*"We take the average runoff in all grid cells intersecting Rio Behar (ranging from 1-8 SMB grid cells for the four models) to arrive at a single hourly runoff value for each SMB model following Smith et al (2017)."*

L161-2. "allowing us to calibrate the free parameters of the routing model (Section 3.2) and to adjust water excess of the SMB models to best match these observations." Suggest delete this as this was explained earlier.
*Amended as requested*

L174/5. I don't quite follow this sentence. Do you mean "Our overall goal for this study is to improve current understanding of supraglacial hydrological transport processes through application of a hydrological model, which includes classical hillslope and channel routing processes, to a catchment on the GrIS" ?
*That is precisely what we mean. We now state "Our overall goal for this study is to improve current understanding of supraglacial hydrological transport processes by classically modelling hillslope and channel routing"*

L193. " Figure 1 shows an overall schematic of our approach." This sentence comes too late as Fig 1 has already been referred to. Could just delete or move to after first sentence of this section on line 175.
*Deleted as requested*

L195/6 "we are interested in generalizing the process of water routing from satellite image collection to water routing" doesn't make sense to me.
*It didn't make sense to Reviewer 1 either! It now states "we are interested in generalizing the process of water routing in cases where pre-existing channel network maps do not exist. Further, we must generate different river networks to test the effects of network density on the routing model."*

L199-203. I don't quite follow these two sentences. Could they say "Having done

this, conventional network generation was confounded due to two large topographic depressions, one located in the upper part of the catchment and one located near the catchment outlet. Standard DEM preparation for hydrological analysis (in which the upstream depression was filled while the outlet depression was preserved) generated unrealistic parallel drainage channels upstream and no channels in the outlet depression of the catchment" ?

*Reviewer 1 was similarly confused by this passage, so it has been substantially reorganized and re written. The passage now stands as*

[revised manuscript text omitted]

L215/16. I don't follow this sentence. Should this say " ...we model the networks as static snapshots within HRR, which is..." ? I assume networks should be plural as there is the coarse one and the fine one. Need to explain what you mean by "loosely coupled to SMB runoff" or delete this.
*This sentence now reads "Instead, we model this these network snapshots with HRR as loosely coupled to SMB runoff (as opposed to tightly coupled, where SMB runoff would be an input into network generation), which is reasonable for our one-month experiment (Section 3.3.1)."*

L233-5. Sentence here seems misplaced. Better to justify using HRR earlier, when you first introduce it - i.e. Section 1.
*Since this is a hydrology model, we wanted to remind glaciology readers about why we chose it here, as we felt the detail in the preceding section about network extraction might have reasonably distracted someone from remembering it if given earlier.*

L278-80. This sentence doesn't quite makes sense. Suggest rewrite to clarify.
*We have added the following here, which links to a few sentences later "(e.g. fine networks with hillslope processing take much longer to run and therefore used less generations, see below)"*

L282. What is a "population member"? This is not explained.
*This refers to a population size of 40. We had introduced the concept of population size, which is equivalent to n population members, but we see the confusion. We replaced this with " we used a population size of 40"*

L282/3. On L175 you refer to "two experimental settings (inclusion/exclusion of hillslope flow, coarse/fine channel network densities)". See also nomenclature used in Fig 1 e.g. "hillslope coarse" or "non hillslope fine". Can you use consistent terminology so it's clear that you're running tests that both include and exclude hillslope processes and also that the 'coarse' and 'fine' refer to the network density? The terms you use here "non-hillslope tests", "coarse hillslope test" and "fine hillslope test" are a bit sloppy and

rather confusing and don't match earlier. See also legend in Fig 3.

*Good idea. We have added to the L175 passage "These runs are labelled as either 'fine/coarse' and 'hillslope/non-hillslope', so for example an experiment using a fine network density and excluding hillslope processes would be labelled 'non-hillslope coarse.'" This harmonizes Fig 1, 3, and this passage of text.*

L315-18. Confusion over whether you're defining routing delay to be generic or specifically in terms of comparison of peaks. Would it be better to say something like: "Finally, we calculate routing delays for each of our 16 calibrated routing models. Routing delay will be a function of both time of day and discharge but here we calculate routing delay as the difference in ADCP peak and the unrouted SMB runoff peak. This delay is the shortest for MERRA2 (1-3 hours) and longest for MAR and RACMO2 (5-6 hours). These values provide an estimate for daily peak flow delay between runoff forcing and the calibrated HRR model."

*This is a great suggestion! We've rewritten along the lines of your suggestion as: "Finally, we calculate routing delays for each of our 16 calibrated routing models by noting the difference in ADCP peak and the unrouted SMB runoff peak. Routing delay is a function of both time of day and discharge, but is easiest to interpret at daily peak flow. This peak delay is the shortest for MERRA2 (1-3 hours) and longest for MAR and RACMO2 (5-6 hours). This routing delay is a function of both time of day and discharge, so tThese values represent an approximate estimate for daily peak flow delay between runoff forcing and calibrated HRR model, and. These calculations represent the total travel time for water to pass through the system, from runoff production to the outlet. "*

L366/7. You say "across inclusion/exclusion of hillslope process and across coarse/fine networks". So this nomenclature matches initial terminology (line 175) but not that used in other places – see comment re L282/3

*Amended here and before as noted.*

Fig 7 Heading. You say "Mean" but these are box whisker plots so explain what all features represent. What are the dots?

*We see the confusion here. These are standard boxplots- median drawn as a black line, IQR given as colored shading, and dots represent outliers. The 'mean' refers to the fact that we take the mean Manning's n of all rivers in the bin, and then make a boxplot of those mean values across the experimental controls. So, the first boxplot has 8 values in it- one mean n for all streams in that bin for the 8 hillslope tests.*

*The caption now reads "Figure 7. Mean Manning's n for all rivers binned by area, where bin refers to an area threshold given in Table 1. Bins are bounded by the maximum value indicated in the x axes and a minimum value equal to the maximum area of the next smallest bin. There are eight values per each boxplot: these represent the mean Manning's n for all channels in that area bin for each of eight experimental trial configurations. Our experiment design yields for instance eight models that include hillslopes (four of which are coarse, and four of which are fine), and these boxplots plot the mean n, per bin, of those eight models. Boxplots are standard and show median, IQR, and outliers. Non-hillslope trials require substantially more friction than hillslope trials in the largest channels, suggesting compensation for lack of hillslope process representation."*

In the body of the paper, can you explain the reason for the very anomalous results for
bin 3.16 km and 10 km where Manning's n is very high?
*This suggests to us that the model chose to suddenly add quite a bit of friction to these large channels
(note the quite high friction of bin >10 as well) in the absence of hillslopes to slow water otherwise in
order to conserve mass.*

*This is now acknowledged as "Non-hillslope large channels in the three highest orders require a
substantially larger Manning's n value than these same channels with a hillslope process included,
indicating that the non-hillslope models necessitate higher friction in large channels to match outlet
flows. For the 2nd and 3rd largest bins, this resulted in extreme friction in those channels just before the
basin outlet in order to provide enough friction to conserve mass and momentum."*

L382. Should 'and/or' just be 'and'? The runs that included hillslope processes and
had a fine channel network were best right?
*Not uniformly the best. The top performing model used a coarse network with hillslope process, so we
wanted this and/or language to reflect that.*

L387. In addition to ref. Karlstrom and Yang, 2016 you could also include:
Koziol, C., Arnold, N., Pope, A. and Colgan, W., 2017. Quantifying supraglacial meltwater
pathways in the Paakitsoq region, West Greenland. Journal of Glaciology,
as they too model channel incision.
In addition to Yang et al, 2020 you could also refer to:
Banwell, A., Hewitt, I., Willis, I. and Arnold, N., 2016. Moulin density controls drainage
development beneath the Greenland ice sheet. Journal of Geophysical Research Earth
Surface, v. 121, p.2248-2269. and Koziol, C.P. and Arnold, N., 2018. Modelling seasonal meltwater
forcing of the velocity of land-terminating margins of the Greenland Ice Sheet. The Cryosphere, v. 12,
p.971- 991 as they too couple SMB models to surface and subglacial routing models to examine
basal water pressures.
*Cited as recommended.*

L398-405. You find very slow water transport rates on hillslopes here and suggest this
may be due to slow transport through an ice crust. But what role does ignoring travel
through lakes esp. while lakes may be filling play on your results?
*Per the larger discussion above, we are confident that lakes are treated effectively, if not explicitly.*

L427. You say ": : :parameters should be able to accurately appliable to flow route
watering in: : :" This is not grammatically correct. Also, what is 'watering'? I'm not sure
this statement is correct is it? Don't you imagine there is a huge evolution on some of
these parameter values over the summer?
*Well this sentence was certainly garbled! Our apologies for not thoroughly catching it.  This now reads
"While we have here only reported flows during a verifiable 72-hour period, in theory our model
parameters should be able to accurate route water in similar areas of the GrIS with similar network
drainage patterns in similar seasons."*

L434-5. Regarding this sentence, lakes may be filling over this period which would explain
the discrepancy. Could it also be due to leakage into the ice sheet via crevasses?

*Both of these are plausible reasons, but we unfortunately cannot verify either. Please see our earlier response for how we have handled this valid concern.*
* * *
L29/30. ': : :explicitly including hillslope flow and routing runoff through a realistically fine channel network: : :' It's difficult to fully understand this without reading the rest of the paper. Could you somehow add "as opposed to not including hillslope processes" and "as opposed to a course channel network"?
L38. 'with unique and complex hydrologic process distinct from terrestrial hydrology' This is rather convoluted. Could it just be changed to 'which is distinct from those in terrestrial settings'?
L27. 'routed to match measured flows' could be deleted as it's repetition.
L48. Should 'ice' be changed to "glacier" or "ice sheet" or "snow, firn and ice"?
L80. ': : :at the global: : :'
L81. Is 'paradigm' a little grand? Would "approach" be better instead?
L90. ": : :a model such as: : :"
L90 change "science" to "understanding" ?
L93. Suggest change 'are interacting" to "interact".
L97/8. I think the semicolons would be better as periods.
L120. Suggest change ": : :basin: we here: : :" to ": : :basin. Here we: : :"
L115. Delete "(Section 2.2) as you're referring to Smith et al 2017 here.
L122. Delete "as"
L124. Consider changing ": : :catchment. However: : :" to ": : :catchment, but the..."
L134. Would ": : :2 m resolution portion of the ArcticDEM was obtained..." sound better than ": : :2 m resolution ArcticDEM DEM was obtained : : :"?
L134 "The ArcticDEM has been: : :"
L154. Suggest ": : :hydrology and its discharge values are frequently: : :"
L154. Suggest ": : :(Gleason and Durand, 2020). Further reading: : :"
L157. Suggest changing "and ending" to "to".
L192. "framework to guide"
L204 delete "Rio Behar"
L205 "ultimately"
L212-214. Would it be better to say: "It would be possible to derive expected rates of incision (and additional meltwater supply) due to frictional heating of the channels, but without including a radiation budget and ice property data we could not model how: : :"
L217. Could delete "ultimately"
L219. Suggest ": : :higher order) and the fine network had seven: : :"
L232. ": : :Section 3.2 is required: : :"
L349. ": : :.large data sets: : :"
L350. "Figure 5"
* * *
*Amended as suggested. Thanks for taking the time to document these thoughtful language edits! They are hard for us to see (having stared at the paper for far too long) and thus these really improve this manuscript.*

---

## Author Response (AR1)

This manuscript was revised exactly according to our response to reviewers- all of our changes are documented, line by line, in those documents.

In addition, we have included discussion of two additional papers at the request of the editor.

---

## Author Response (AR2)

We have made all of the editor's suggested changes.

In addition, while preparing our files for archival and reproduction, we noticed a numerical error in Figure 4. The original Figure 4 is at left, and the corrected version is at right. In the corrected version, we see the basic results still hold: three of the models vastly overpredict runoff without routing. However, the corrected version shows that the total mass passing the watershed is even closer to observed than previously reported.

This discrepancy arose from initially counting the total volume of the SMB models throughout the entire time domain rather than limiting to just the 72hour validation period. The ratio of instantaneous to routed runoff is identical and unaffected, and no other results are affected- this was purely an error of visualization that has been corrected as part of our cross check before final publication file uploads. All text pertaining to Figure 4 has been updated accordingly. The only sentences that need be changed are as follows

L367 Despite indicating that reduced input runoff is required to route flows accurately across all models, overall routed cumulative discharge was lower than in situ measurements for this time period for coarse networks due to underprediction of low flows, and overpredicted using fine networks (Figure 4).

Fig 4 caption- Calibrated models underpredict water export due to underestimation of night-time low flows for coarse networks, and overpredict total water export with fine networks.

[Figure]

We have cross checked all other results and find no errors, and see this as a strength of the archival process, even if we are embarrassed.